# Gaussian Herding across Pens: An Optimal Transport Perspective on Global Gaussian Reduction for 3DGS

**Tao Wang**[1*]   **Mengyu Li**[2*]   **Geduo Zeng**[1]   **Cheng Meng**[13†]   **Qiong Zhang**[13†]

[1]Center for Applied Statistics, Institute of Statistics and Big Data, Renmin University of China
[2] Department of Statistics and Data Science, Tsinghua University
[3] Big Data and Responsible Artificial Intelligence for National Governance, Renmin University of China
wang_tao@ruc.edu.cn   mengyuli@tsinghua.edu.cn   geduozeng@ruc.edu.cn
chengmeng@ruc.edu.cn   qiong.zhang@ruc.edu.cn

## Abstract

3D Gaussian Splatting (3DGS) has emerged as a powerful technique for radiance field rendering, but it typically requires millions of redundant Gaussian primitives, overwhelming memory and rendering budgets. Existing compaction approaches address this by pruning Gaussians based on heuristic importance scores, without global fidelity guarantee. To bridge this gap, we propose a novel optimal transport perspective that casts 3DGS compaction as global Gaussian mixture reduction. Specifically, we first minimize the composite transport divergence over a KD-tree partition to produce a compact geometric representation, and then decouple appearance from geometry by fine-tuning color and opacity attributes with far fewer Gaussian primitives. Experiments on benchmark datasets show that our method (i) yields negligible loss in rendering quality (PSNR, SSIM, LPIPS) compared to vanilla 3DGS with only 10% Gaussians; and (ii) consistently outperforms state-of-the-art 3DGS compaction techniques. Notably, our method is applicable to any stage of vanilla or accelerated 3DGS pipelines, providing an efficient and agnostic pathway to lightweight neural rendering. The code is publicly available at `https://github.com/DrunkenPoet/GHAP`

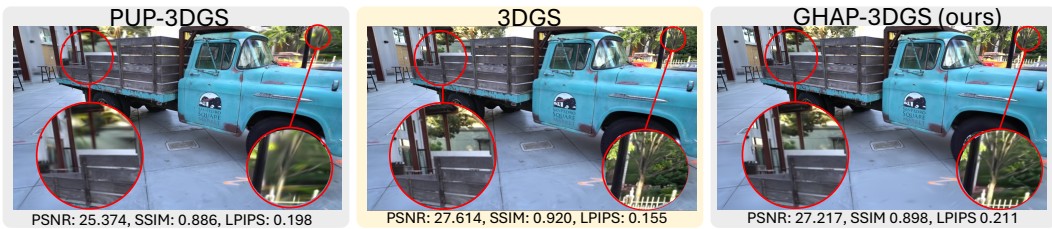

Figure 1: **Visual comparison**. When reducing the number of Gaussians by 90%, our method outperforms other compaction techniques, such as PUP-3DGS, and remains competitive with the original 3DGS.

## 1   Introduction

Real-time 3D scene reconstruction and rendering dynamically generates photorealistic 3D representations from sensor data (e.g., multi-view images, LiDAR) with minimal latency, enabling critical applications in augmented/virtual reality (AR/VR), autonomous navigation, and immersive media [1, 2]. The current state-of-the-art, 3D Gaussian Splatting (3DGS) [3], iteratively learns 3D

39th Conference on Neural Information Processing Systems (NeurIPS 2025).

anisotropic Gaussian primitives with color and opacity attributes to model these scenes. During rendering, these 3D Gaussians are projected to 2D screens and $\alpha$-blended to achieve real-time photorealistic synthesis.

However, 3DGS faces significant efficiency challenges: its iterative densification process often produces millions of redundant Gaussians for complex scenes [4, 5, 6]. This inefficiency leads to high memory/storage costs and increased per-frame rendering time, limiting deployment on resource-constrained platforms like mobile and AR/VR devices [7].

A common strategy to improve efficiency is compaction [8, 9, 10, 7]reducing the number of Gaussians while preserving rendering fidelity. Fewer primitives result in reduced storage needs and faster rendering, improving per-frame performance. This approach is viable because 3DGS densification inherently generates redundant primitives [4, 5, 6]. Existing methods achieve compaction via pruning or random subset selection [11, 12,

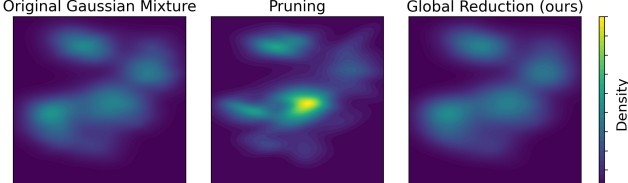

Figure 2: **Comparison of heuristic pruning and our method**. The original mixture (left) with $10^3$ components is reduced to 5% using either pruning (middle) or our method (right). Our method better preserves the overall structure.

5, 13, 14, 15, 6, 4, 16]. These strategies naïvely discard Gaussian primitives; while simple to implement, they are often ineffective. In particular, they tend to lose critical structural details (as shown in Fig. 1) or distort the underlying geometry (as shown in Fig. 2). Such losses can degrade rendering quality, particularly in regions with fine-scale features or complex material properties. These limitations motivate our key question:

> *How to design an efficient compaction method that preserves 3D spatial and structural geometry?*

To address this, we frame compaction as an optimization problem, where the goal is to approximate the original 3DGS representation with fewer Gaussians. Our solution leverages a statistical perspective, treating the scene's geometric structure as a probabilistic model and employing principled reduction techniques to preserve fidelity.

(i) **Geometric compaction via GMR.** We first observe that the geometry of a 3DGS representationdefined by the positions, covariances, and opacities of its Gaussian primitivescan be interpreted as a Gaussian mixture model (GMM). Here, the mixture density is a convex combination of individual Gaussian densities, weighted by their opacities. This formulation naturally connects 3DGS compaction to Gaussian Mixture Reduction (GMR), a well-studied problem in statistics where a high-order GMM is approximated by one with fewer components while minimizing a divergence measure.

(ii) **Appearance optimization.** While geometry is compacted via GMR, the appearance of the scenegoverned by the color attributes of the Gaussiansmust also be preserved. To achieve this, we decouple the optimization of geometry (position, covariance) and appearance (color, opacity). After GMR-based compaction, we fine-tune the color and opacity of the reduced set of Gaussians using the standard 3DGS training pipeline. This two-stage strategy ensures that the compacted model maintains both geometric accuracy and photorealistic rendering quality.

For GMR, we minimize the composite transportation divergence [17], which is rooted in optimal transport theory [18, 19] and allows an effective algorithm. We tailor this GMR algorithm (detailed in Section 3.2.1) so that it scales well in scenes like 3DGS with an extensive amount of Gaussians. Crucially, our GMR optimizer does not merely select a subset of existing Gaussians; instead, it creates new primitives that can dynamically adjust their positions and covariances to better approximate the underlying geometry (as shown in Figure 2). Our method is fundamentally algorithm-agnostic: it functions as a plug-and-play module that can enhance both the standard 3DGS pipeline and any of its variants (Section 4), applicable at any stage of training to boost computational efficiency.

To summarize, our contributions are:

- We open a new pathway to view 3DGS representations as a Gaussian mixture and perform compaction from the perspective of Gaussian mixture reduction via optimal transport. This contrasts with prior compaction methods that ignore geometric structure and often produce distortions, whereas our approach preserves geometric fidelity, offering a new and impactful direction for 3DGS.

- We are the first to adapt GMR to 3DGS. We introduce a novel cost function that yields closed-form, low-cost updates. We also develop a block-wise GMR algorithm guided by a KD-tree, enabling efficient largescale scene compaction. These strategies are non-trivial and bridges theory with practical scalability.

- Our method is post-hoc and compatible with any existing 3DGS pipeline, making it highly practical and broadly applicable. With minimal overhead, our approach achieves SOTA compaction performance, both in quality and efficiency.

- Empirical results demonstrate that our method preserves rendering quality at 10% retention ratio.

## 2 Related Works

**Compaction Techniques in 3DGS.** 3DGS compaction seeks to minimize the Gaussian count while preserving image quality. Existing work falls into two operations: densification (where to add) and pruning (what to drop). Most current approaches rely on per-Gaussian heuristic scores.

For densification, Taming3DGS [10] ranks candidate Gaussians by combining gradient, pixel coverage, per-view saliency, and core attributes including opacity, depth, and scale; Color-cued GS [20] considers the view-independent spherical harmonics coefficient gradient to better capture color cues; and GaussianPro [9] guides growth using depth and normal maps. In the pruning phase, LightGaussian [11], Mini-Splatting [12], RadSplat [5], and AtomGS [13] compute an importance score from each Gaussian's accumulated ray contribution, typically a mix of volume, opacity, transmittance, and hit count, and discard the lowest-ranked Gaussians. Gradient-aware variants prune by per-Gaussian gradients (Trimming-the-Fat) [14] or the second-order sensitivity score derived from the Hessian matrix (PUP-3DGS) [15]. The score can also be trained via a learnable mask, as in LP-3DGS [6], Compact3DGS [4], and HAC [16]. Moreover, multi-view consistency criteria discard Gaussians unseen by keyframes [21] or visible only in real but not virtual views [22]. For a comprehensive survey of 3DGS compression and compaction, we refer readers to [8, 23].

Despite their success, most existing strategies evaluate each Gaussian independently, leaving open the question of whether the retained set is truly the best global surrogate. Our work addresses this gap from a probabilistic perspective via Gaussian mixture reduction.

**3DGS from Probabilistic Distribution Point of View.** Kheradmand et al. [24] formulate 3DGS as a Markov chain Monte Carlo process and use stochastic gradient Langevin dynamics to migrate dropped Gaussians onto retained ones, partially recycling lost information. However, their update remains pairwise and lacks convergence guarantees under a principled divergence. Moreover, this scheme is coupled with the original 3DGS pipeline, limiting its generalizability to other variants.

## 3 Method: Gaussian Herding across Pens

### 3.1 Probabilistic Scene Representation

Let $\phi(x; \mu, \Sigma) = |2\pi\Sigma|^{-1} \exp(-(x - \mu)^\top \Sigma^{-1}(x - \mu))$ be the PDF of a Gaussian distribution with mean $\mu$ and covariance $\Sigma$. A Gaussian mixture with $n$ components is a distribution with density:

$$\phi_n(x) = \sum_{i=1}^{n} \alpha_i \phi(x; \mu_i, \Sigma_i),$$

where $\alpha_i$ are the mixture weights satisfying $\alpha_i > 0$[1]. In the context of 3DGS, let $x \in \mathbb{R}^3$ be spatial coordinates and $f(x)$ represent the implicit surface function (*i.e.,* geometric shape and opacity that

---

[1]In statistics, Gaussian mixtures are defined with the constraint $\sum_{i=1}^{n} \alpha_i = 1$ to ensure the area under the density function is 1. However, we relax this constraint and consider unnormalized Gaussian mixtures, as in 3DGS, where the integral under the geometric surface need not equal 1.

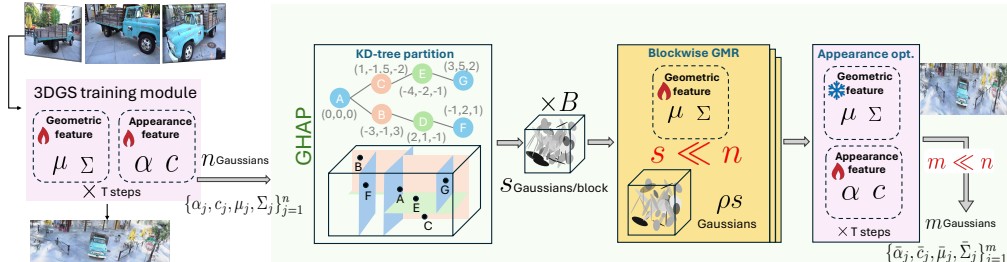

Figure 3: **An illustration of the proposed GHAP approach**. The process begins with full-resolution 3DGS training to obtain initial geometric and appearance features. These Gaussians are then spatially partitioned using a KD-tree and grouped into blocks–analogous to **sheep pens**. We then perform blockwise Gaussian Mixture Reduction (GMR) to approximate the geometric shape within each block using a much smaller number of Gaussians. This step is analogous to the popular **kernel herding** method [25]. Finally, a lightweight appearance refinement step further optimizes the appearance feature of the reduced set. This multi-stage pipeline progressively guides the Gaussians in each block–analogous to **herding across pens**–toward a compact and high-fidelity representation.

excludes color). The training process in 3DGS learns opacity parameter $\alpha_i$, location parameter $\mu_i$, and shape parameter $\Sigma_i$ such that

$$\phi_n(x) \approx f(x), \quad \forall x \in \mathcal{X},$$

where $\mathcal{X}$ is the 3D scene volume. Therefore, the *geometry* of the 3D scene can be effectively represented by a Gaussian mixture. Then, each of these Gaussian primitive is associated with its own color $c_i$. Both the geometry and appearance attributes are important for high quality rendering.

## 3.2 Compaction via Optimal Transport

Motivated by the observation that many 3DGS algorithms [4, 5, 6] produce a significant number of redundant Gaussians during training, we improve rendering efficiency through compactionreducing the number of Gaussian primitives to achieve lower memory usage and faster rendering while preserving visual fidelity. The process consists of two key phases:

1. **Geometric Compaction via GMR**: Leveraging our probabilistic interpretation, we formulate compaction as Gaussian Mixture Reduction (GMR) [26], approximating the original Gaussian mixture with redundant components by one with fewer components. This yields a compacted geometric representation: $\bar{\phi}_m(x) = \sum_{j=1}^{m} \bar{\alpha}_j \phi(x; \bar{\mu}_j, \bar{\Sigma}_j)$, where $m \ll n$. This step modifies only the Gaussian positions ($\bar{\mu}_j$) and covariances ($\bar{\Sigma}_j$), leaving appearance attributes unchanged.

2. **Appearance Optimization**: The reduced Gaussians are initialized with appearance attributes (colors, opacities) and fine-tuned for optimal rendering performance. This step optimizes appearance only, maintaining geometric consistency.

Our approach decouples geometry and appearance optimization while using standard 3DGS training to preserve quality. Our training pipeline is visualized in Figure 3 and we describe the details below.

### 3.2.1 Geometric Compaction via GMR

Following Zhang et al. [17], we formulate compaction as minimizing the composite transportation divergence (CTD) between two Gaussian mixtures:

**Definition 1** (Composite transportation divergence). *Let $c(\cdot, \cdot)$ be a divergence between two Gaussians. The composite transportation divergence (CTD) between two Gaussian mixtures $\phi_n(x) = \sum_{i=1}^{n} \alpha_i \phi(x; \mu_i, \Sigma_i)$ and $\phi'_m(x) = \sum_{j=1}^{m} \alpha'_j \phi(x; \mu'_j, \Sigma'_j)$ with cost function $c(\cdot, \cdot)$ is*

$$\mathcal{T}_c(\phi_n, \phi'_m) = \inf \left\{ \sum_{i=1}^{n} \sum_{j=1}^{m} \pi_{ij} c(\phi(\cdot; \mu_i, \Sigma_i), \phi(\cdot; \mu'_j, \Sigma'_j)) : \sum_{j=1}^{m} \pi_{ij} = \alpha_i, \sum_{i=1}^{n} \pi_{ij} = \alpha'_j \right\}.$$

The CTD generalizes *optimal transport* [18] to mixtures, treating each component as a discrete distribution in the space of Gaussian distributions. The cost function measures the cost of moving one unit of Gaussian from one location to another, and $\pi_{ij}$ measures the corresponding amount of mass that is being moved. The total cost is proportional to the cost and the mass, and the divergence is the smallest transportation cost to move the original mixture to the target mixture. The reduced mixture becomes

$$\{\bar{\alpha}_j, \bar{\mu}_j, \bar{\Sigma}_j\} = \underset{\{\alpha'_j, \mu'_j, \Sigma'_j\}}{\arg\min} \, \mathcal{T}_c(\phi_n, \phi'_m). \tag{1}$$

With this formulation, the Gaussian mixture after compaction has optimal guarantee. The solution also guides the choice of $m$ to balance compactness and fidelity.

As shown in Zhang et al. [17], (1) can be solved using the effective iterative algorithm in Algorithm 1.

The algorithm reduces to a $k$-means variant in Gaussian space: 1) The assignment step follows the same principle as traditional $k$-means, but replaces the $L^2$ distance between vectors with a cost function $c(\cdot, \cdot)$ between Gaussian distributions. 2) The update step generalizes the cluster center computation: In traditional $k$-means, centers are updated as arithmetic averages (barycenters w.r.t. $L^2$ distance) of vectors in each cluster. In this algorithm, centers become barycenters of Gaussians in each cluster, minimized w.r.t. cost function $c(\cdot, \cdot)$. Thus, standard $k$-means emerges as a special case when using $L^2$ distance on vectorized Gaussian parameters.

While the standard GMR algorithm provides optimal theoretical guarantees, its direct application to 3DGS compaction proves computationally prohibitive. Although the algorithm must converge in finite steps [17][2], the assignment step involves $nm$ evaluations of the cost function

---

**Algorithm 1** GMR via $k$-means Clustering

1: Initialize $\{\bar{\mu}_j^{(0)}, \bar{\Sigma}_j^{(0)}\}_{j=1}^m$
2: **for** t=1,..., **do**
3:     *Assignment Step:*
4:     **for** $i = 1$ to $n$ **do**         ▷ $\mathcal{O}(nm)$
5:         Assign component $i$ to cluster $\mathcal{C}_j$ that minimizes $c(\phi(\cdot; \mu_i, \Sigma_i), \phi(\cdot; \bar{\mu}_j^{(t-1)}, \bar{\Sigma}_j^{(t-1)}))$
6:     **end for**
7:     *Update Step:*
8:     **for** $j = 1$ to $m$ **do**         ▷ $\mathcal{O}(nm)$
9:         Compute new cluster center: $\bar{\mu}_j^{(t)}, \bar{\Sigma}_j^{(t)} = \arg\min \sum_{i \in \mathcal{C}_j} \alpha_i c(\phi(\cdot, \mu_i, \Sigma_i), \phi(\cdot, \mu, \Sigma))$
10:     **end for**
11:     **if** no change in assignments **then**
12:         **for** $j = 1$ to $m$ **do**     ▷ $\mathcal{O}(n)$
13:             Compute weight: $\bar{\alpha}_j = \sum_{i \in \mathcal{C}_j} \alpha_i$
14:         **end for**
15:         break
16:     **end if**
17: **end for**

---

per iteration. In typical 3D scenes, the number of Gaussians scales as $n = \Omega(10^5)$[3], and even after 95% reduction, each iteration would still require at least $10^8$ operations and memory storage. For the update step, the computational cost for the cluster center depends on the pre-specified cost function $c(\cdot, \cdot)$. The KL divergence considered in Zhang et al. [17] suffers from (a) significant overhead of computing $O(\rho s^2 \log n)$ covariance matrices inversions, and (b) numerical instability due to small eigenvalues of covariance matrices. To overcome this challenge, we introduce two key optimizations designed for 3DGS:

- **Blockwise GMR via KD-Tree**: To improve computational and memory efficiency during training, we partition the scene into spatially blocks and perform GMR within each block. As demonstrated in Remark 1, this blockwise approach yields significant computational savings. While both KD-trees [27] and Octrees [28] are effective for spatial partitioning in 3D space, we employ a KD-tree for two key advantages. First, it produces more balanced partitions across regions. Second, it avoids unnecessary subdivisions in sparse regions that would waste computational resources.

  Our KD-tree is constructed solely from Gaussian centers $\{\mu_i\}_{i=1}^n$ (justified by the observed small eigenvalues of covariance matrices). Each split uses the median coordinate value, creating $2^d$ blocks at depth $d$. We set $d = \lfloor \log_2(n/s) \rfloor$ to ensure blocks contain at most $s$ Gaussians with $s \ll n$, then reduce each block to $m = \rho s$ components ($\rho$ = retention ratio).

  **Remark 1** (Computational Cost Comparison). *Our blockwise approach reduces the per-iteration computational cost from $O(\rho n^2)$ to $O(\rho s^2)$ per block.[4] With $2^{depth} = O(\log n)$ blocks in total, the*

---

[2]In particular, we find that in our experiment, the algorithm converges in only around 6 iterations.
[3]The $a_n = \Omega(b_n)$ $(O(b_n))$ if there exist $C \geq 0$ such that $a_n \geq C b_n$ $(a_n \leq C b_n)$.
[4]Each block reduces from $s$ Gaussian components to $\rho s$.

*overall complexity becomes $O(\rho s^2 \log n)$. For typical values of $n = 10^5$, $s = 10^3$ and $\rho = 0.05$, this reduces the cost from $10^8$ to approximately $10^5$ operations–a substantial improvement. The savings become even more pronounced for larger $n$. Furthermore, the reduction steps can be executed in parallel across blocks, offering additional computational speedup.*

- **Efficient Cost Function**: We introduce a novel cost function that overcomes the limitations of the KL divergence used in [17] by being computationally efficient without sacrificing approximation quality. Our proposed divergence is:

$$c(\phi(\cdot; \mu, \Sigma), \phi(\cdot; \mu', \Sigma')) = \|\mu - \mu'\|_2^2 + \|\Sigma - \Sigma'\|_F^2, \tag{2}$$

which offers three significant advantages: First, it preserves distributional similarity, as Gaussian distributions are uniquely determined by their mean and covariance. Second, the assignment step requires only efficient vector and matrix norm computations. Third, the update step simplifies to calculating weighted averages, thereby avoiding the computationally expensive covariance matrix inversions required by the KL divergence:

$$\bar{\mu}_j^{(t)} = \frac{\sum_{i \in \mathcal{C}_j} \alpha_i \mu_i}{\sum_{i \in \mathcal{C}_j} \alpha_i}, \qquad \bar{\Sigma}_j^{(t)} = \frac{\sum_{i \in \mathcal{C}_j} \alpha_i \Sigma_i}{\sum_{i \in \mathcal{C}_j} \alpha_i}.$$

### 3.2.2 Appearance Optimization

Following geometric compaction, we initialize the appearance attributes (opacity and color) of the compacted Gaussian primitives. For each primitive in the reduced mixture, we assign the appearance parameters from its closest counterpart in the original Gaussian mixture. Using these initial values, we then optimize the appearance attributes through backpropagation within the standard 3DGS training pipeline used in the first stage. We optimize the opacity instead of directly using the values from the GMR algorithm because its output weights do not necessarily satisfy the constraint that opacity must be between 0 and 1. Fine-tuning the opacity leads to better visualization performance.

---

**Algorithm 2** `GHAP`: 3DGS Compaction via Block-wise GMR

---

1: **Input:** Trained 3DGS model for $T$ steps to obtain $\{(\alpha_i, \mu_i, \Sigma_i, c_i)\}_{i=1}^n$, retention ratio $\rho$
2: **Output:** Compacted $\{(\bar{\alpha}_j, \bar{\mu}_j, \bar{\Sigma}_j, \bar{c}_j)\}_{j=1}^m$
3: **Stage 1: Geometric Compaction**
4: 1.  Build KD-tree from Gaussians $\{\mu_i\}_{i=1}^n$ with depth $d = \lfloor \log_2(n/s) \rfloor$ ⊳ $\mathcal{O}(nd \log n)$
5: 2. For each leaf block $\mathcal{B}_k$: ⊳ $\mathcal{O}(nmT/2^d)$
6:    Run Algorithm 1 to reduce to $\rho s$ Gaussians
7: **Stage 2: Appearance Optimization**
8: 1. Initialize appearance for each $\bar{\phi}_j$:
9:    $\bar{c}_j \leftarrow c_{i^*}$ and $\bar{\alpha}_j \leftarrow \alpha_{i^*}$ where $i^* = \arg\min_{i \in [n]} \|\mu_i - \bar{\mu}_j\|_2$
10: 2.  Fine-tune $\{\bar{\alpha}_j, \bar{c}_j\}$ using standard 3DGS rendering pipeline for $T$ steps

---

### 3.3 Training Details with `GHAP` Algorithm

Integrating these components, we present our complete training pipeline in Algorithm 2, called Gaussian Herding Across Pens (`GHAP`). The process begins with standard 3DGS optimization for $T$ steps, followed by blockwise GMR. We then freeze the geometric parameters ($\mu$ and $\Sigma$) while fine-tuning the appearance attributes (opacity $\alpha$ and color $c$) through an additional $T$-step 3DGS optimization. This procedure can be applied iteratively throughout training as needed.

## 4 Experiments

### 4.1 Experimental Setup

**Datasets.** For a comprehensive evaluation of `GHAP` algorithm, we use three real-world datasets: *Tanks & Temples* [29], *Mip-NeRF 360* [30], and *Deep Blending* [31], which cover varying levels of detail, lighting conditions, and scene complexities. For each dataset, we adopt the same scenes as in [8].

- **Tanks & Temples**: We evaluate two unbounded outdoor scenes, "Truck" and "Train", both featuring centered viewpoints.
- **Mip-NeRF 360**: We test on a mix of indoor and outdoor scenes, including "Bicycle", "Bonsai", "Counter", "Flowers", "Garden", "Kitchen", "Room", "Stump", and "Treehill", all with centered viewpoints.

- **Deep Blending**: We include two indoor scenes, "Dr. Johnson" and "Playroom", where the viewpoint is directed outward.

**Baselines.** To evaluate the effectiveness of our proposed method, we compared it against four strong compaction techniques: **LightGaussian** [11], **PUP-3DGS** [15], **Trimming the Fat** [14], and **MesonGS** [32], as well as four end-to-end 3DGS variants: **Mini-Splatting(-D)** [12], **AtomGS** [13], **3DGS-MCMC** [24], and **LocoGS** [33]. Notably, Mini-Splatting-D and AtomGS were employed as backbone models in our approach, while the other variants were used for direct comparative evaluation against the compaction methods. A consistent evaluation protocol was established to ensure fair and reliable conclusions. For post-training compaction methods applicable to pre-trained models–including GHAP, LightGaussian, PUP-3DGS, Trimming the Fat, and MesonGS–we initialized all from the same backbone model (trained using vanilla 3DGS, Mini-Splatting-D, or AtomGS for 15k iterations) and applied their respective compaction procedures directly, excluding any compression-specific modules. All models subsequently underwent identical fine-tuning for 15k iterations to achieve the target retention ratio. For the other end-to-end variants (MiniSplatting, 3DGS-MCMC and LocoGS), we executed training for 30k iterations under their default configurations. Detailed experimental steps for each method can be found in the appendix.

**Evaluation Metrics.** We assess 3DGS compaction using standard metrics for rendering quality. We report: (1) **PSNR**, measuring pixel-level accuracy; (2) **SSIM**, evaluating perceptual similarity based on luminance, contrast, and structure; and (3) **LPIPS**, capturing perceptual distance via a learned model. Higher PSNR/SSIM and lower LPIPS indicate better quality. For each method, we also report the corresponding **number of Gaussian primitives**.

## 4.2 Quantitative Results

We first show that our approach outperforms other SOTA compaction techniques. Second, we demonstrate the effectiveness of our GHAP compaction method as a plug-in module within 3DGS and its variants. We present the experimental results followed by our key findings.

**Comparison with SOTA.** We compare our method against a comprehensive set of baselines, which can be categorized into two groups:

- **End-to-End Compact 3DGS Variants:** Mini-Splatting [12], 3DGS-MCMC [24], and LocoGS [33]. Note that for LocoGS, the final number of Gaussians is not a user-controllable parameter.

- **Post-Training Compaction Baselines:** LightGaussian [11], PUP-3DGS [15], Trimming the Fat [14], and MesonGS [32]. These methods all use a standard 3DGS backbone and apply pruning-based techniques for compaction.

Our approach is evaluated in two configurations: **3DGS+GHAP** and **MiniSplatting+GHAP**. The former uses the same vanilla 3DGS backbone and 10% retention rate as the pruning baselines for a direct comparison of compaction strategies. The latter replaces the built-in pruning step in Mini-Splatting with our GHAP algorithm to demonstrate its effectiveness on a different backbone.

Quantitative results are summarized in Table 1, with methods grouped by their final primitive count. As expected, LocoGS achieves strong performance due to its larger primitive count. Among methods with comparable primitive counts (Group 2), our **3DGS+GHAP** achieves superior performance in SSIM and PSNR, with a marginally lower LPIPS score. The advantage of our compaction strategy is further evident when using the Mini-Splatting backbone. Our **MiniSplatting+GHAP** outperforms other compaction-based approaches while often using fewer primitives. As shown in Fig. 4 (left), this performance lead is consistent across a wide range of retention ratios, not just at $\rho = 0.1$.

**Runtime & Memory Usage Comparison.** Crucially, the improved performance of our method does not come at a computational cost. As depicted in Fig. 4 (middle), our method's runtime is faster than all baselines except the exceptionally swift Trimming the Fat. While our method exhibits a slightly higher memory footprint (in Fig. 4 right) during compaction due to pairwise distance computation in each KD-tree block, the difference is not substantial (less than an order of magnitude).

**As a Plug-In Compaction Method.** Our method can be used as a plug-in compaction method in various 3DGS training algorithms. This demonstrates the broad applicability of our proposed method. To verify this, we apply our compaction method within various 3DGS pipelines. Specifically, we

Table 1: **Quantitative comparison of rendering quality against SOTA methods**. Our method (GHAP), applied to two different backbones (3DGS and Mini-Splatting), consistently matches or surpasses pruning-based baselines, even at a low retention rate (10%), while maintaining competitive runtime and memory usage. The best results for a given number of Gaussians are shown in **bold**; second best are underlined.

| Method | Tanks&Temples | | | | MipNeRF-360 | | | | Deep Blending | | | |
|---|---|---|---|---|---|---|---|---|---|---|---|---|
| | SSIM↑ | PSNR↑ | LPIPS↓ | $k$ Gaussians | SSIM↑ | PSNR↑ | LPIPS↓ | $k$ Gaussians | SSIM↑ | PSNR↑ | LPIPS↓ | $k$ Gaussians |
| Vanilla 3DGS | 0.853 | 23.785 | 0.169 | 1577 | 0.813 | 27.554 | 0.221 | 2627 | 0.907 | 29.816 | 0.238 | 2475 |
| LocoGS | 0.843 | 23.655 | 0.191 | 571 | 0.798 | 27.049 | 0.257 | 674 | 0.903 | 29.972 | 0.261 | 529 |
| 3DGS+GHAP (ours) | **0.818** | **23.312** | 0.242 | 157 | 0.764 | **26.404** | 0.314 | 263 | **0.905** | **29.647** | **0.264** | 248 |
| LightGaussian | 0.756 | 22.113 | 0.306 | 158 | 0.735 | 25.674 | 0.331 | 263 | 0.869 | 28.010 | 0.327 | 248 |
| PUP-3DGS | 0.767 | 21.519 | 0.280 | 158 | 0.753 | 25.332 | 0.309 | 262 | 0.895 | 29.153 | 0.274 | 248 |
| Trimming the Fat | 0.776 | 21.535 | 0.293 | 156 | 0.731 | 25.255 | 0.343 | 263 | 0.887 | 28.056 | 0.302 | 247 |
| MesonGS | 0.811 | 20.714 | **0.208** | 157 | **0.773** | 24.924 | **0.264** | 263 | 0.896 | 28.693 | **0.264** | 248 |
| 3DGS-MCMC | 0.779 | 22.141 | 0.282 | 157 | 0.763 | 25.957 | 0.309 | 263 | 0.885 | 28.976 | 0.298 | 248 |
| MiniSplatting | 0.799 | 22.661 | 0.265 | 78 | 0.759 | 26.022 | 0.318 | 111 | 0.895 | 29.395 | 0.289 | 125 |
| MiniSplatting+GHAP (ours) | **0.835** | **23.232** | **0.198** | 79 | **0.802** | **27.090** | **0.250** | 112 | **0.909** | **30.042** | **0.254** | 127 |

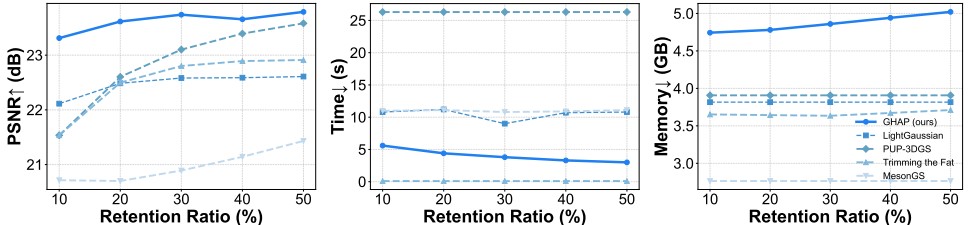

Figure 4: **Comparison of compaction methods on the Tanks & Temples dataset**. Left: Rate-Distortion (RD) curves; middle: computational time, and right: memory consumption.

consider three representative variants as the backbone: 3DGS [3] , AtomGS [13], and Mini-Splatting-D [12]. Each of them employs a distinct densification strategy, and our method can be directly embedded into the pipeline without extensive engineering effort. For each backbone, we evaluate performance under two retention ratios (10% and 20%). Tab. 2 summarizes our experimental results.

Table 2: **Quantitative results with different backbones**. The compaction performance of our `GHAP` method when used with different 3DGS variants as backbones under varying retention ratios.

| Backbone | $\rho$ | Tanks&Temples | | | | MipNeRF-360 | | | | Deep Blending | | | |
|---|---|---|---|---|---|---|---|---|---|---|---|---|---|
| | | SSIM↑ | PSNR↑ | LPIPS↓ | $k$ Gaussians | SSIM↑ | PSNR↑ | LPIPS↓ | $k$ Gaussians | SSIM↑ | PSNR↑ | LPIPS↓ | $k$ Gaussians |
| 3DGS-30k | 10% | 0.818 | 23.312 | 0.242 | 157 | 0.764 | 26.404 | 0.314 | 263 | 0.905 | 29.647 | 0.264 | 248 |
| | 20% | 0.835 | 23.615 | 0.212 | 314 | 0.788 | 26.973 | 0.275 | 526 | 0.907 | 29.864 | 0.252 | 596 |
| | 100% | 0.853 | 23.785 | 0.169 | 1577 | 0.813 | 27.554 | 0.221 | 2627 | 0.907 | 29.816 | 0.238 | 2475 |
| Mini-Splatting-D | 10% | 0.835 | 23.232 | 0.198 | 313 | 0.802 | 27.090 | 0.250 | 357 | 0.909 | 30.042 | 0.254 | 331 |
| | 20% | 0.855 | 23.403 | 0.171 | 626 | 0.821 | 27.310 | 0.214 | 614 | 0.912 | 30.170 | 0.238 | 662 |
| | 100% | 0.848 | 23.338 | 0.140 | 3132 | 0.832 | 27.486 | 0.176 | 3578 | 0.907 | 29.980 | 0.211 | 3316 |
| AtomGS | 10% | 0.793 | 22.988 | 0.274 | 189 | 0.764 | 26.535 | 0.307 | 293 | 0.899 | 29.347 | 0.282 | 271 |
| | 20% | 0.812 | 23.282 | 0.240 | 378 | 0.788 | 27.025 | 0.269 | 586 | 0.902 | 29.347 | 0.268 | 542 |
| | 100% | 0.814 | 23.289 | 0.235 | 1897 | 0.796 | 27.135 | 0.251 | 2928 | 0.902 | 29.314 | 0.267 | 2709 |

The quantitative results in Tab. 2 demonstrate that our method effectively preserves the backbone models' visual quality, even at an extreme retention rate of 10%. Notably, on some scenes (highlighted in bold), the compacted model's performance surpasses that of the uncompacted backbone.

Beyond quantitative metrics, we provide a qualitative analysis by visualizing multiple scenes before and after compaction in Fig. 5. As evidenced in the figure, our method successfully preserves rendering quality across most scenes while using only 10% of the Gaussian primitives. Interestingly, in certain cases, our compaction not only preserves but *surpasses* the original quality. A representative example is the "Kitchen" scene (Mini-Splatting-D backbone). We conjecture this improvement occurs because Mini-Splatting-D lacks a pruning mechanism, often generating an over saturated

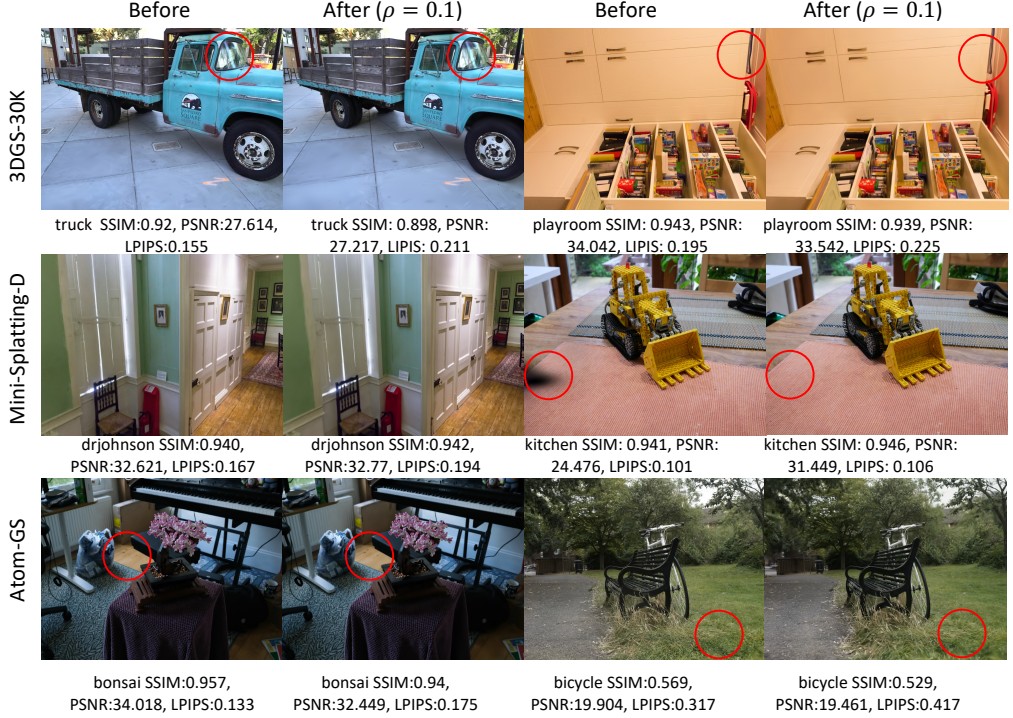

Figure 5: **Visual Quality Before and After Compaction.** Visual results for various scenes under different 3DGS backbones, compacted to 10% of their primitives using our GHAP method. Our approach preserves rendering quality with negligible loss. In some cases (e.g., "Kitchen"), compaction even improves quality by regularizing an over saturated Gaussian distribution.

set of Gaussians that introduce visual artifacts (*e.g.*, the unnatural shadows in the lower-left region). Our method acts as a global regularizer, mitigating this issue by reducing unnecessary density while improving the overall expressiveness and preserving the underlying 3D structure. Naturally, our approach is inherently limited by the quality of its input. If the original model suffers from significant artifacts due to a lack of primitives in certain regions, our compaction cannot resolve these fundamental issues. This limitation is demonstrated in the "Bicycle" scene, where artifacts present in the Atom-GS backbone persist after compaction.

### 4.3  Ablation Studies

**Influence of KD Tree Depth.** To validate the effectiveness of the KD-tree partitioning strategy, we provide an ablation study on it. Due to the large scale of the three primary datasets, low KD-tree depths result in an excessive number of points per block, making it infeasible to run the GMR algorithm. Therefore, we conduct ablation experiments on the smaller mic scene from NeRF-Synthetic. Results in Fig. 6 show that increasing KD-tree depth reduces memory usage and runtime, while PSNR first improves and then declines. This indicates that moderately finer partitions allow GMR to compact regions more effectively, whereas overly fine splits may fragment primitives and degrade quality.

**Loss Function Design.** As shown in Tables 1 and 2, our method exhibits a slight underperformance on the LPIPS metric. To address this, we investigate whether incorporating an LPIPS loss term can yield improvements.

Our baseline loss function follows the vanilla 3DGS formulation, using an L1-to-SSIM ratio of 8:2. We experiment with two new weighting schemes that include LPIPS: (1) L1:SSIM:LPIPS = 8:1:1, and (2) L1:SSIM:LPIPS = 6:2:2. The results of this ablation study on the Tanks&Temples dataset are shown in Fig. 7 (with additional results in the Appendix). Our analysis reveals a trade-off between perceptual and distortion metrics: increasing the weight of the LPIPS loss improves LPIPS scores but leads to a slight deduction in PSNR and SSIM.

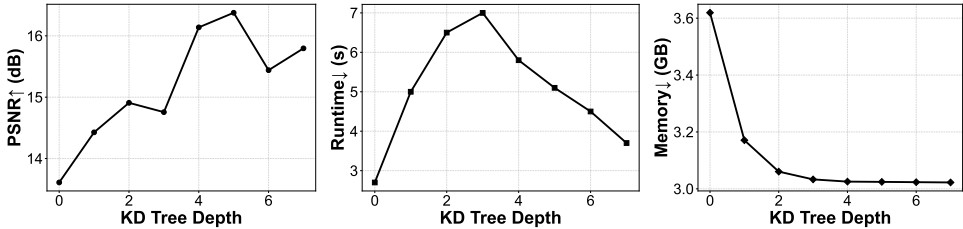

Figure 6: **Hyperparameters**. The impact of KD tree depth on runtime performance, memory footprint, and rendering quality.

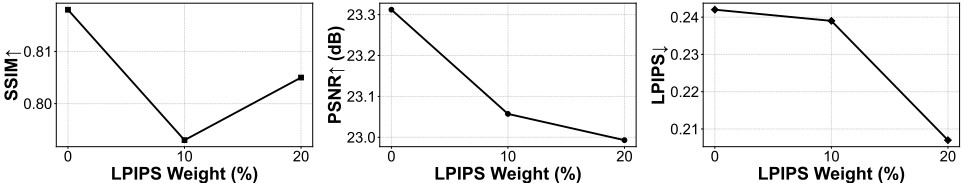

Figure 7: **Comparison of different loss weights**. A trade-off between perceptual and distortion metrics.

**GMR versus Random Subsampling.** We conduct an ablation study on the Tanks&Temples dataset to evaluate the contribution of each stage in our pipeline: geometric compaction and appearance optimization. A random subsampling baseline is included to assess the effectiveness of our design choices.

The results, presented in Table 3, compare two compaction schemes followed by the same fine-tuning procedure. Our findings demonstrate that: first, our compaction procedure is significantly more effective than random subsampling and other pruning baselines (as shown in Tab. 1). Second, the subsequent appearance optimization stage provides substantial quantitative improvements for both compaction approaches. The significant performance gain over the baseline validates the necessity and effectiveness of both stages in our proposed pipeline.

Table 3: **Ablation study**. Mean PSNR, SSIM, and LPIPS on the Tanks and Temples dataset at each stage of `GHAP` pipeline, with random subsampling as a control. Each operation is applied cumulatively to all subsequent stages.

| Method | Tanks&Temples | | |
|---|---|---|---|
| | SSIM↑ | PSNR↑ | LPIPS↓ |
| 3DGS-15K | 0.839 | 23.084 | 0.194 |
| + 10% `GHAP` Compaction | 0.502 | 14.015 | 0.483 |
| + 10% Random Compaction | 0.333 | 9.573 | 0.555 |
| +15K Fine-tuning | 0.818 | 23.312 | 0.242 |
| +15K Fine-tuning | 0.712 | 21.312 | 0.282 |

## 5   Conclusion and Discussion

We propose an optimal transport-based Gaussian mixture reduction framework for 3D Gaussian Splatting, achieving compact yet faithful representations. By minimizing composite transport divergence with appearance fine-tuning, our method preserves high visual fidelity while retaining only 10% of Gaussians, outperforming prior compaction techniques. The framework scales efficiently via block-wise KD-tree partitioning and integrates seamlessly with diverse 3DGS pipelines.

Future directions include enhancing robustness across challenging scene types, incorporating perceptual objectives, developing multi-scale and overlap-aware partitioning, adopting auto-tuned schedules, and extending to dynamic 3DGS for real-time temporal rendering.

## Acknowledgments and Disclosure of Funding

Qiong Zhang is supported by the National Natural Science Foundation of China Grant 12301391. Cheng Meng and Qiong Zhang are supported by Big Data and Responsible Artificial Intelligence for National Governance, Renmin University of China. The authors would like to thank the anonymous reviewers for their constructive suggestions.

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

# Appendix

# A    More Details about the Blockwise GMR Compaction

## A.1    Introduction to Optimal Transport (OT)

The Optimal Transport (OT) problem dates back to 1781, when French mathematician *Gaspard Monge (1746–1818)* first formulated it as finding the minimal-cost way to move sand into a hole. This "transport plan" minimizes the transportation cost, hence the term "optimal transport". The original formulation is known as Monge's problem. In 1947, Russian economist *Leonid Kantorovich (1912–1986)* relaxed Monge's formulation, leading to the so-called Monge-Kantorovich problem.

Specifically, let $P$ and $Q$ be two distributions over a metric space $\mathcal{X}$, let $c : \mathcal{X} \times \mathcal{X} \to \mathbb{R}_+$ be a cost function, and let the coupling

$$\Pi(P, Q) = \left\{ \pi(x, y) : \int \pi(dx, \cdot) = Q(\cdot), \int \pi(\cdot, dy) = P(\cdot) \right\}$$

be the set of joint distributions with marginals $P$ and $Q$. For any cost function $c$, the total transportation cost induced by a plan $\pi \in \Pi(P, Q)$ is defined as

$$\mathcal{I}_c(\pi) = \int_{\mathcal{X} \times \mathcal{X}} c(x, y) \, \pi(dx, dy).$$

Here, $\pi(x, y)$ indicates how much "mass" is transported from $x$ to $y$. The first constraint, $\int \pi(x, dy) = P(x)$, ensures that the mass at location $x$ is spread over $\mathcal{X}$, while the second constraint, $\int \pi(dx, y) = Q(y)$, ensures that the destination at $y$ receives the required mass. The OT problem seeks the optimal plan

$$\pi^* = \arg\min \left\{ \mathcal{I}_c(\pi) : \pi \in \Pi(P, Q) \right\},$$

minimizing the transportation cost. The corresponding minimal cost,

$$\mathcal{T}_c(P, Q) = \mathcal{I}_c(\pi^*),$$

induces a divergence between $P$ and $Q$. This OT divergence provides a principled way to measure distributional similarity, enabling applications in density matching, generative modeling, and dimensionality reduction.

The optimal plan $\pi^*$ and OT divergence typically lack closed-form solutions. Numerical algorithms [18] are used to compute the OT between discrete measures. When $P$ and $Q$ are discrete, the OT divergence reduces to:

**Example 1** (OT Divergence Between Discrete Distributions). *For* $P = \sum_{i=1}^{n} u_n \delta_{x_n}$ *and* $Q = \sum_{j=1}^{m} v_m \delta_{y_m}$, *the OT divergence is*

$$\mathcal{T}_c(P, Q) = \min \left\{ \sum_{i=1}^{n} \sum_{j=1}^{m} \pi_{ij} c(x_i, y_j) : \sum_{i=1}^{n} \pi_{ij} = v_j, \sum_{j=1}^{m} \pi_{ij} = u_i \right\}. \tag{3}$$

Exact solutions can be found via linear programming, while approximations are obtained using algorithms like Sinkhorn.

## A.2 Relationship Between Composite Transportation Divergence and OT

The Composite Transportation Divergence (CTD) [34, 35] extends OT to Gaussian mixtures. For two mixtures $\phi_n = \sum_{i=1}^{n} \alpha_i \phi(\cdot; \mu_i, \Sigma_i)$ and $\phi'_m = \sum_{j=1}^{m} \alpha'_j \phi(\cdot; \mu'_j, \Sigma'_j)$, the CTD is:

$$\mathcal{T}_c(\phi_n, \phi'_m) = \inf \left\{ \sum_{i=1}^{n} \sum_{j=1}^{m} \pi_{ij} c(\phi(\cdot; \mu_i, \Sigma_i), \phi(\cdot; \mu'_j, \Sigma'_j)) : \sum_{j=1}^{m} \pi_{ij} = \alpha_i, \sum_{i=1}^{n} \pi_{ij} = \alpha'_j \right\},$$

Comparing this with the discrete OT divergence (3), the CTD treats Gaussian mixtures as discrete distributions over the space of Gaussians, defining the divergence as the OT between them.

To illustrate, consider $n$ warehouses and $m$ factories in the space of Gaussian distributions $\mathcal{F}$. The $i$th warehouse, at $\phi(\cdot; \mu_i, \Sigma_i)$, holds $\alpha_i$ units of material, while the $j$th factory, at $\phi(\cdot; \mu'_j, \Sigma'_j)$, requires $\alpha'_j$ units. The cost to transport material from $i$ to $j$ is $c(\phi(\cdot; \mu_i, \Sigma_i), \phi(\cdot; \mu'_j, \Sigma'_j))$, and $\pi_{ij} \geq 0$ denotes the transported amount. The total cost under plan $\pi$ is $\sum_{i,j} \pi_{ij} c(\phi(\cdot; \mu_i, \Sigma_i), \phi(\cdot; \mu'_j, \Sigma'_j))$. The coupling set $\Pi(\alpha, \alpha') = \{\pi_{ij} : \sum_{j=1}^{m} \pi_{ij} = \alpha_i, \sum_{i=1}^{n} \pi_{ij} = \alpha'_j\}$ ensures: (a) correct material removal from warehouses, and (b) correct delivery to factories. The OT problem seeks the plan $\pi^* \in \Pi(\alpha, \alpha')$ minimizing the total cost. The minimal cost corresponds to the CTD between the mixtures, quantifying the optimal transport cost between them.

Our compaction method leverages this interpretation, approximating a mixture with fewer Gaussians by minimizing their CTD-based dissimilarity.

## A.3 Algorithm Intuition

At first glance, the optimization problem in (1) is bilevel: the OT plan must be found for any candidate $\{\bar{\alpha}_j, \bar{\mu}_j, \bar{\Sigma}_j\}_{j=1}^{m}$, and the objective must be minimized. However, as shown by Zhang et al. [17], this simplifies with a clear interpretation. The column-wise marginal constraints on $\pi$ are redundant, and each Gaussian primitive $(\mu_i, \Sigma_i)$ transports its full mass to its "closest" counterpart. The optimal plan thus corresponds to clustering $\{(\mu_i, \Sigma_i)\}_{i=1}^{n}$ into $m$ clusters $\{\mathcal{C}_j\}_{j=1}^{m}$, with the original Gaussians forming cluster "barycenters". Mathematically, the simplified program is:

$$\min \left\{ \sum_{j=1}^{m} \sum_{i \in \mathcal{C}_j} \pi_{ij} c(\phi(\cdot; \mu_i, \Sigma_i), \phi(\cdot; \bar{\mu}_j, \bar{\Sigma}_j)) : [n] = \mathcal{C}_1 \sqcup \cdots \sqcup \mathcal{C}_m \right\},$$

where $\sqcup$ denotes disjoint union. The optimal plan has a closed form:

$$\pi_{ij} = \begin{cases} \alpha_i & \text{if } j = \arg\min_k c\left(\phi(\cdot; \mu_i, \Sigma_i), \phi(\cdot; \bar{\mu}_k, \bar{\Sigma}_k)\right), \\ 0 & \text{otherwise.} \end{cases}$$

With this, $\pi$ and the reduced mixture parameters can be updated alternately, formalized in the following $k$-means-like algorithm:

- **Assignment Step:** Each Gaussian $\phi(\cdot; \mu_i, \Sigma_i)$ is assigned to cluster $\mathcal{C}_j$ by minimizing $c(\phi(\cdot; \mu_i, \Sigma_i), \phi(\cdot; \bar{\mu}_j, \bar{\Sigma}_j))$, analogous to $k$-means' nearest-centroid assignment.
- **Update Step:** For the cost function

$$c(\phi, \phi') = \|\mu - \bar{\mu}\|_2^2 + \|\Sigma - \bar{\Sigma}\|_F^2,$$

the parameters $(\bar{\mu}_j, \bar{\Sigma}_j)$ are updated as weighted averages:

$$\bar{\mu}_j = \frac{\sum_{i \in \mathcal{C}_j} \alpha_i \mu_i}{\sum_{i \in \mathcal{C}_j} \alpha_i}, \quad \bar{\Sigma}_j = \frac{\sum_{i \in \mathcal{C}_j} \alpha_i \Sigma_i}{\sum_{i \in \mathcal{C}_j} \alpha_i}.$$

This mirrors $k$-means' centroid update.

## A.4 Algorithm Convergence

The following theorem guarantees convergence of the CTD sequence generated by Algorithm 1. In the worst case, the algorithm requires exponentially many steps, but it typically converges in about 5 iterations. A full proof is given in Zhang et al. [17].

**Theorem 1** (Convergence of the Algorithm). *Suppose $c(\cdot, \cdot)$ is continuous, and for any $\Delta > 0$ and $\phi^\star$, the set $\{\phi : c(\phi^*, \phi) \leq \Delta\}$ is compact under some Gaussian space metric. Let $\{\bar{\phi}_m^{(t)}\}$ be the sequence generated by the update step with initial $\{\bar{\alpha}_j^{(0)}, \bar{\mu}_j^{(0)}, \bar{\Sigma}_j^{(0)}\}$, and let $\mathcal{T}_c^{(t+1)}$ be the CTD at iteration $t$. Then:*

1. *There exists $T$ and $\{\bar{\alpha}_j^*, \bar{\mu}_j^*, \bar{\Sigma}_j^*\}$ such that for all $t \geq T$, $\{\bar{\alpha}_j^{(t)}, \bar{\mu}_j^{(t)}, \bar{\Sigma}_j^{(t)}\} = \{\bar{\alpha}_j^*, \bar{\mu}_j^*, \bar{\Sigma}_j^*\}$ and $\mathcal{T}_c^{(t+1)} = \mathcal{T}_c^{(*)}$, where $\mathcal{T}_c^{(*)}$ is the CTD between the original mixture and $\{\bar{\alpha}_j^*, \bar{\mu}_j^*, \bar{\Sigma}_j^*\}$.*

2. *The limit point $\{\bar{\alpha}_j^*, \bar{\mu}_j^*, \bar{\Sigma}_j^*\}$ is a local minimum of $\mathcal{T}_c$.*

3. *An MM-based exhaustive algorithm with $O(m^n)$ complexity solves (1).*

# B Experiment Steps and Complexity Analysis in More Detail

## B.1 Detailed 3DGS Pipline

The advantages of 3DGS in rendering speed and image fidelity have made it applicable to a wide range of tasks, including human reconstruction, AI-generated content, autonomous driving, and beyond [36, 37, 38, 39]. Extensions to dynamic 3DGS, editable 3DGS, and surface representation have further broadened its utility [40, 41, 42]. 3D Gaussian Splatting (3DGS) represents a scene as a set of anisotropic 3D Gaussian primitives, each parameterized by its spatial location, shape, opacity, and radiance. The pipeline consists of the following key steps:

1. **Initialization**. From Structure-from-Motion (SfM), obtain calibrated camera poses and a sparse point cloud. Each point is initialized as a 3D Gaussian with an opacity $\alpha_i$:

$$\phi(x; \mu_i, \Sigma_i) = |2\pi\Sigma|^{-1} \exp\left(-\frac{1}{2}(x - \mu_i)^T \Sigma_i^{-1}(x - \mu_i)\right),$$

   where $\mu_i \in \mathbb{R}^3$ is the position (mean), $\Sigma_i \in \mathbb{R}^{3 \times 3}$ is the covariance matrix (anisotropic shape), and $\alpha_i \in [0, 1)$.

2. **Projection & Rasterization**. Each 3D Gaussian is projected to 2D using the camera model,

$$\Sigma_i' = JW\Sigma_i JW^T,$$

   where $W$ is a veiw transformation matrix and $J$ is the Jacobian of the projective transform. Rasterization is done using a differentiable splatting approach which makes optimization possible.

3. **Image Formation (Alpha Blending)**. The pixel color is computed via volumetric blending:

$$C = \sum_{i=1}^{N} T_i \cdot \alpha_i \cdot c_i \text{ with } T_i = \prod_{j=1}^{i-1}(1 - \alpha_j),$$

   where $c_i$ is the SH-predicted color of the $i$-th Gaussian in front-to-back order.

4. **Optimization**. The Gaussian parameters $\{\mu_i, \Sigma_i, \alpha_i, c_i\}$ are optimized to minimize a photometric loss:

$$\mathcal{L} = (1 - \lambda)\left\|\widehat{C} - C^\star\right\|_1 + \lambda \cdot \mathcal{L}_{\text{SSIM}},$$

   where $\widehat{C}$ is the rendered image, $C^\star$ the ground truth, and $\lambda \in [0, 1]$ balances the two loss terms.

5. **Adaptive Densification & Pruning**. During training, Gaussians are cloned (under-reconstruction) or split (over-reconstruction) based on view-space gradient magnitude, and low-contribution Gaussians are pruned.

## B.2 Algorithm Complexity

For Algorithm 1, we discuss its computational cost for reducing $n$ Gaussians to $m$ Gaussians. The assignment step requires computing pairwise distances between all $n$ input Gaussians and $m$ cluster centers, resulting in a complexity of $\mathcal{O}(nm)$. The update step involves computing the barycenters. Since the assignments are already known and the update for each cluster is linear in the number of assigned Gaussians, this step has a complexity of $\mathcal{O}(n)$. The overall time complexity of Algorithm 1 is $\mathcal{O}(nm)$.

Our `GHAP` algorithm consists of two components: *Geometric Compaction* and *Appearance Optimization* (fine-tuning). In this analysis, we focus only on the time complexity of the geometric compaction stage. The KD-tree is constructed by recursively splitting the dataset along the median of a selected coordinate axis until a maximum depth $d = \lfloor \log_2(n/s) \rfloor$ is reached. Let the input size be $n$, and the time complexity of building the tree be $T(n)$. This follows the recurrence: $T(n) = 2T(n/2) + \mathcal{O}(n \log_2 n)$. The recursion terminates after $d$ levels, yielding a total KD-tree construction complexity of $\mathcal{O}(nd \log_2(n))$. In the second step of geometric compaction, Algorithm 1 is applied independently within each KD-tree block with $s$ Gaussians reduced to $\rho s$ Gaussians. Based on our analysis in the previous paragraph, the per-block cost is $O(\rho s^2)$. This along with the fact that there are $2^d = n/s$ blocks, the total complexity becomes: $\mathcal{O}\left((n/s) \cdot \rho s^2\right) = \mathcal{O}\left(\rho ns\right) = \mathcal{O}(ms)$. Combining with the KD-tree construction cost, the total time complexity of the geometric compaction step is $\mathcal{O}\left(n(d \log n + mT/2^d)\right)$.

## B.3 Detailed Experiments Steps

To ensure a fair comparison in our experiments, all methods undergo 30,000 total iterations under consistent training conditions. MCMC uses 30K iterations of its own update process in [24]. For other baselines, we summarize the backbone architecture, compaction methods, and fine-tuning steps in the table below.

| Method | Backbone (0-15k iterations) | Compaction (15001th iteration) | Fine-tune (150001-30K iterations) |
|---|---|---|---|
| 3DGS | a | None | d |
| 3DGS+GHAP | a | Our compaction | d |
| LightGaussian | a | LightGaussian compaction | d |
| PUP-3DGS | a | PUP-3DGS compaction | d |
| Trimming the FAT | a | Trimming the FAT compaction | d |
| MesonGS | a | MesonGS compaction | d |
| MiniSplatting | b | MiniSplatting compaction | d |
| MiniSplatting-D+GHAP b | b | Our compaction | d |
| LocoGS | c | LocoGS compaction | d |

a Vanilla 3DGS densification and pruning in [3].

b Mini-Splatting densification and pruning in [12].

c LocoGS 3DGS update in [33].

d 3DGS fine-tuning in [3].

**Key implementation details:**

- For compaction methods applicable to pre-trained models (e.g., LightGaussian, PUP-3DGS, Trimming the FAT, MesonGS), we initialize from the same backbone model (trained using vanilla 3DGS) and apply their respective compaction directlyexcluding any compression modules.

- All methods undergo identical fine-tuning (15k iterations) to achieve the target retention ratio.

This standardized protocol ensures that performance differences stem solely from the methods themselves, eliminating variations due to training procedures. All experiments are conducted on a server with 256 GB RAM and a 96-core Intel Xeon Platinum 8255C CPU, and on a workstation equipped with five NVIDIA RTX 3090 GPUs, each with 24 GB of VRAM.

## C  An Additional Comparison Experiments

As documented in prior work, 3DGS-MCMC [24] reinterpretes the 3D Gaussian Splatting (3DGS) process through the lens of Markov Chain Monte Carlo (MCMC), thereby naturally exploring a broader parameter space. In particular, it formulates pruning as a state transition within the MCMC framework and incorporates $L_1$ regularization, avoiding abrupt hard-threshold deletions and enabling the natural removal of unnecessary Gaussian elements. Given these properties, a further comparative analysis with 3DGS-MCMC is warranted.

To facilitate a fair comparison, we configured 3DGS-MCMC under two settings: one constrained to 300k Gaussians, and another initialized with 3000k Gaussians followed by compaction using our method to reduce the count to 300k. Quantitative results on Tandt & Temples are summarized in the table below. As evidenced by the results presented in the table, GHAP combined with 3DGS-MCMC

| Method | SSIM↑ | PSNR↑ | LPIPS↓ |
|---|---|---|---|
| 3DGS-MCMC-300k | 0.813 | 22.925 | 0.239 |
| 3DGS-MCMC-3000k+GHAP | 0.827 | 22.786 | 0.209 |

yields superior performance compared to the native 3DGS-MCMC approach constrained to 300k Gaussians. This demonstrates the effectiveness of our method in accurately approximating the surface geometry of 3DGS, highlighting the advantage of our compaction strategy.

## D  Ablation on Joint Geometry and Appearance Fine-tuning

In our configured compaction process, compaction was performed only once at the 15,001st iteration. We conducted an ablation study on the timing of compaction to examine the impact of varying compaction frequencies on the final outcome. Two alternative compaction strategies were considered: the first involved compacting to 20% at the 15,001st iteration, followed by an additional 50% compaction at the 20,001st iteration; the second strategy applied compaction of 40%, 50%, and 50% at the 15,001st, 20,001st, and 25,001st iterations, respectively. All three strategies ultimately resulted in a final retention rate of 10%. Experimental results, as presented in the table below, indicate that jointly refining geometry and appearance over multiple stages does not lead to evidently improved performance. Therefore, in practical applications, emphasis should be placed on executing appearance optimization for as long as possible, rather than pursuing multi-stage compaction.

| Compacting Iteration | $\rho$ | SSIM↑ | PSNR↑ | LPIPS↓ |
|---|---|---|---|---|
| 15001 | 0.1 | 0.817 | 23.313 | 0.242 |
| 15001, 20001 | 0.2,0.5 | 0.817 | 23.44 | 0.247 |
| 15001, 20001, 25001 | 0.4,0.5,0.5 | 0.811 | 23.359 | 0.255 |

## E  Additional Numerical Results and Scene Visualizations

We report a more comprehensive set of results for the comparison experiments in Table 4, including one different retention ratio: 20%. As shown in the updated results, our compaction method significantly outperforms other methods either post-processing compression methods or end-to-end methods.

In addition to the previously shown Figure 5, we present more detailed qualitative comparisons across multiple scenes in Figure 8. Our method consistently preserves the visual quality of the original models. In particular, when applied to stronger 3DGS variants with improved densification strategies, such as Mini-Splatting-D, our compaction framework performs even better. This is reflected in the fact that, after compaction, Mini-Splatting-D often achieves higher rendering quality than the original 3DGS baseline.

Table 4: We compared GHAP with four post-processing methods (LightGaussian, PUP-3DGS, Trimming the Fat, MesonGS) at 20% retention, as well as two end-to-end methods (LocoGS, 3DGS-MCMC). GHAP also replaces the pruning in Mini-Splatting. Results show that our method substantially outperforms both post-processing and end-to-end approaches.

| Method | Tanks&Temples | | | | MipNeRF-360 | | | | Deep Blending | | | |
|---|---|---|---|---|---|---|---|---|---|---|---|---|
| | SSIM↑ | PSNR↑ | LPIPS↓ | k Gaussians | SSIM↑ | PSNR↑ | LPIPS↓ | k Gaussians | SSIM↑ | PSNR↑ | LPIPS↓ | k Gaussians |
| original 3DGS | 0.853 | 23.785 | 0.169 | 1577 | 0.813 | 27.554 | 0.221 | 2627 | 0.907 | 29.816 | 0.238 | 2475 |
| LocoGS | 0.843 | 23.655 | 0.191 | 571 | 0.798 | 27.049 | 0.257 | 674 | 0.903 | 29.972 | 0.261 | 529 |
| 3DGS+GHAP (ours) | **0.835** | **23.615** | 0.212 | 314 | 0.788 | **26.973** | 0.275 | 527 | **0.907** | **29.864** | 0.252 | 496 |
| LightGaussian-20% | 0.779 | 22.486 | 0.271 | 315 | 0.765 | 26.353 | 0.288 | 526 | 0.873 | 28.011 | 0.315 | 495 |
| PUP-3DGS-20% | 0.809 | 22.603 | 0.228 | 315 | **0.790** | 26.671 | **0.257** | 525 | 0.905 | 29.719 | **0.248** | 495 |
| Trimming the Fat-20% | 0.819 | 22.498 | 0.232 | 315 | 0.781 | 26.494 | 0.280 | 524 | 0.900 | 29.082 | 0.272 | 494 |
| MesonGS-20% | 0.822 | 20.699 | **0.207** | 314 | 0.776 | 25.006 | 0.262 | 527 | 0.897 | 28.696 | 0.262 | 496 |
| 3DGS-MCMC | 0.779 | 22.141 | 0.282 | 315 | 0.763 | 25.957 | 0.309 | 263 | 0.885 | 28.976 | 0.298 | 496 |
| MiniSplatting-20% | 0.824 | 22.953 | 0.223 | 142 | 0.794 | 26.728 | 0.267 | 215 | 0.904 | 29.763 | 0.265 | 240 |
| MiniSplatting+GHAP (ours) | **0.855** | **23.403** | **0.171** | 155 | **0.821** | **27.310** | **0.214** | 219 | **0.912** | **30.170** | **0.238** | 250 |

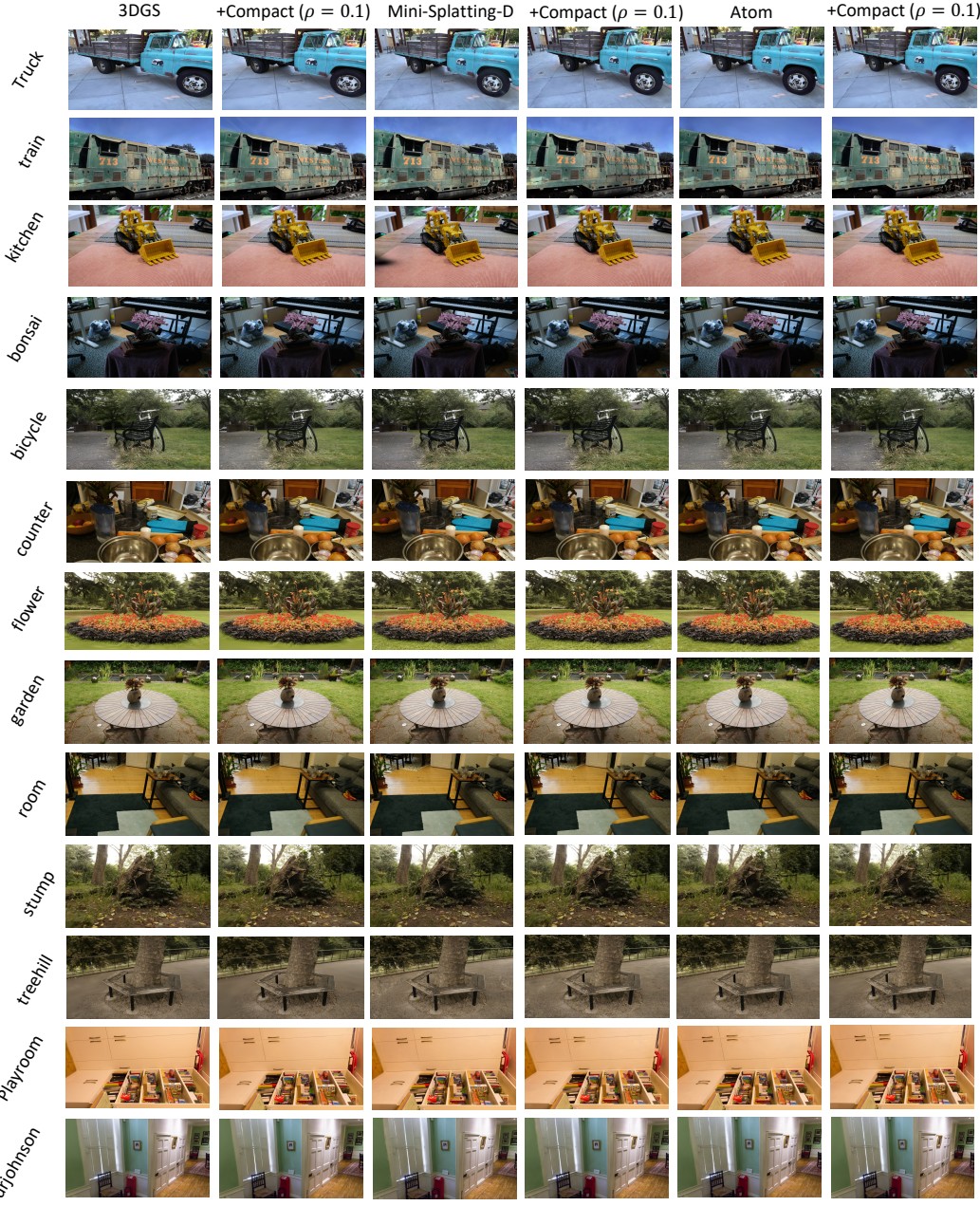

Figure 8: **More scenes visualization**. Visual comparisons across additional scenes before and after compaction.

