# OpenReview forum: "Gaussian Herding across Pens: An Optimal Transport Perspective on Global Gaussian Reduction for 3DGS"
_NeurIPS.cc/2025/Conference — NeurIPS 2025 spotlight_

### Official Review · Reviewer_28Ys · 2025-06-24

**Clarity:** 4
**Significance:** 3
**Originality:** 3
**Rating:** 5
**Confidence:** 4

**Summary:**

The authors propose a compaction method for pre-trained 3DGS scenes that first optimizes a Gaussian mixture model to represent the scene with fewer primitives, followed by a color finetuning that re establishes high quality textures. The method can be applied on top of existing state-of-the-art 3DGS variants for even better rendering quality. As a result, the proposed method achieves high rendering quality, measured with PSNR, SSIM and LPIPS, while using only 10% of the primitive count. To reduce the overall computing time and memory consumption, the scene is first split into blocks using a KD-tree and the GMR is performed on each block separately.

**Questions:**

What is the computing time and memory consumpty of the algorithm?

How does the method compare to Mini-splatting set to a 10% or 20% primitive count?

What can be done to improve the LPIPS quality?

How can you ensure that the settings / loss functions are optimal? Have you made further ablations for different cost functions / fine tuning settings that are not included in the paper?

**Ethical Concerns:**

["NO or VERY MINOR ethics concerns only"]

**Final Justification:**

Given that the authors have addressed all of my concerns (runtime, further comparisons, ablations, code release and formatting) and are willing to apply those changes in the final version, I increase my rating to an accept.

**Limitations:**

The limitations are barely discussed. The authors acknowledge a decrease in LPIPS performance but leave out important information about the runtime and memory consumption.

**Quality:**

3

**Strengths And Weaknesses:**

Strengths:
- The paper is written very well
- The idea is clear and the explanations are easy to follow
- The results are impressive
- The visualizations, algorithms and formulas support the understanding

Weaknesses:
- The additional computing time / memory consumption is not reported. This is the biggest flaw, as it gives no intuition if the proposed method takes minutes, hours or days to compute.
- Comparison to Mini-splatting (set to produce 10% or 20% primitives) is missing. Only Mini-splatting-D is reported
- Line 289: “as demonstrated in the ‘Bicycle’ scene, where the poor performance of the Atom-GS baseline prevents our approach from maintaining”. This does not align with my observation in Figure 4. The grass is clearly better in the “before“ image.
- The random subsampling is barely described and possibly not the best baseline
- The ablation study is very limited. Different cost functions could be compared (for instance: only position, only covariance, additional opacity term)
- Absolute number of Gaussian primitives should be mentioned in the main paper at least once
- Algorithm 2 is not well formatted


Nice to have:
- Experiments that use LPIPS as a loss during appearance optimization
- Source code or example .ply files in the supplementary materials
- Experiments that finetune the geometry as well during appearance optimization
- large Gaussian primitives that overlap with neighboring blocks are not regarded for GMR

---

> ### Author Rebuttal · Authors · 2025-07-31
>
> Thank you for your insightful and positive feedback. We are encouraged that you found the paper well-written, the ideas and explanations clear, and the results impressive. We also appreciate your recognition of the visualizations, algorithms, and formulas in supporting understanding. Below, we address your comments point by point and will incorporate all suggestions into the final version.
>
> - **Run time comparison.** We have included a theoretical analysis in Remark 1 of the manuscript. The total time cost is $O(n\log n + \rho s^2\log n)$. In response to your suggestion, we have also added empirical comparisons of runtime and memory usage against other post-processing compaction baselines (see table below). Our method achieves the **fastest runtime** with **comparable memory consumption**, resulting in **negligible overhead** when integrated into the rendering pipeline—while also delivering the **highest PSNR**.
>
> | Method/$\rho$  | GHAP      |        |        | LightGaussian |        |        | PUP-3DGS  |        |        | Trimming the Fat |        |        | MesonGS   |        |        |
> | :------------- | --------- | ------ | ------ | ------------- | ------ | ------ | --------- | ------ | ------ | ---------------- | ------ | ------ | --------- | ------ | ------ |
> |                | time（s） | memory | PSNR↑  | time（s）      | memory   | PSNR↑  | time（s） | memory | PSNR↑  | time（s）        | memory | PSNR↑  | time（s） | memory | PSNR↑  |
> | 10%            | 5.6       | 4857   | 23.312 | 10.8          | 3908   | 22.113 | 26.3      | 4001   | 21.535 | <0.1             | 3740   | 21.535 | 11.0      | 2830   | 20.714 |
> | 20%            | 4.4       | 4895   | 23.615 | 11.2          | 3908   | 22.486 | 26.3      | 4001   | 22.603 | <0.1             | 3731   | 22.498 | 11.1      | 2830   | 20.699 |
> | 30%            | 3.8       | 4976   | 23.740 | 9.0           | 3908   | 22.582 | 26.3      | 4001   | 23.102 | <0.1             | 3722   | 22.802 | 10.8      | 2830   | 20.891 |
> | 40%            | 3.3       | 5060   | 23.656 | 10.7          | 3908   | 22.588 | 26.3      | 4001   | 23.394 | <0.1             | 3760   | 22.892 | 10.9      | 2830   | 21.142 |
> | 50%            | 3.0       | 5141   | 23.791 | 10.8          | 3908   | 22.609 | 26.3      | 4001   | 23.582 | <0.1             | 3800   | 22.911 | 11.1      | 2830   | 21.432 |
>
> - **Comparison to Mini-splatting.** In line with the your suggestion, we include Mini-Splatting for a more comprehensive comparison. The only difference between MiniSplatting and MiniSplatting-D+GHAP (our GMR pruning integrated into MiniSplatting) lies in the pruning step—our method replaces the default pruning with our proposed GMR-based approach. As shown in the results, this substitution significantly improves performance, thereby **highlighting the effectiveness of our proposed pruning strategy**.
>
> | Method-$\rho$/Dataset         | **TNT**     |              |             |             | **360**     |              |             |             | **DB**      |              |             |             |
> | ----------------------------- |-------------| ------------ | ----------- | ----------- |-------------| ------------ | ----------- | ----------- |-------------| ------------ | ----------- | ----------- |
> |                               | SSIM↑       | PSNR↑        | LPIPS↓      | k Guassians | SSIM↑       | PSNR↑        | LPIPS↓      | k Gaussians | SSIM↑       | PSNR↑        | LPIPS↓      | k Gaussians |
> | MiniSplatting-10%             | 0.799       | 22.661       | 0.265       | 78          | 0.759       | 26.022       | 0.318       | 111         | 0.895       | 29.395       | 0.289       | 125         |
> | MiniSplatting-D+GHAP-10%      | ***0.835*** | ***23.232*** | ***0.198*** | 79          | ***0.802*** | ***27.09***  | ***0.25***  | 112         | ***0.909*** | ***30.042*** | ***0.254*** | 127         |
> | MiniSplatting-20%             | 0.824       | 22.953       | 0.223       | 142         | 0.794       | 26.728       | 0.267       | 215         | 0.904       | 29.763       | 0.265       | 240         |
> | MiniSplatting-D+GHAP-20%      | ***0.855*** | ***23.403*** | ***0.171*** | 155         | ***0.821*** | ***27.31***  | ***0.214*** | 219         | ***0.912*** | ***30.17***  | ***0.238*** | 250         |
>
> - **Design of the loss/cost function and refinement settings.** We would like to clarify that we do not claim our loss function is optimal. The goal of our paper is to show that 3DGS compaction can be framed as a GMR problem. Our proposed loss serves as an initial exploration in this direction and has shown promising results in practice. We conducted additional ablation studies.
>     - **Ablation study on loss function to improve the LPIPS quality.** We evaluated the impact of adding LPIPS to the loss during appearance optimization. As shown below, increasing the LPIPS weight improves LPIPS scores, but slightly reduces PSNR and SSIM:
>
>     | Metric/Loss                          | SSIM↑ | PSNR↑   | LPIPS↓ |
>     | ------------------------------------ | ----- | ------- | ------ |
>     | original (0.8 * L1 + 0.2 * ssim)     | 0.818 | 23.312  | 0.242  |
>     | 0.8 * L1 + 0.1 * ssim +  0.1 * lpips | 0.793 | 23.0572 | 0.239  |
>     | 0.6 * L1 + 0.2 * ssim + 0.2 * lpips  | 0.805 | 22.993  | 0.207  |
>
>     - **Ablation study on other cost functions.** Since a Gaussian primitive includes both mean and covariance, considering only position or covariance during compaction is insufficient. Per your suggestion, we tested adding opacity to the cost function. However, this yielded no notable improvement:
>
>     | Metric/Cost   | SSIM↑         | PSNR↑    | LPIPS↓   |
>     |---------------|---------------|----------|----------|
>     | original cost | 0.818         | 23.312   | 0.242    |
>     | + opacity     | 0.814         | 23.266   | 0.250    |
>
>
> - **Ablation study on joint geometry and appearance fine tuning.** We also tested multi-stage compaction during appearance optimization. As shown below, jointly refining geometry and appearance over multiple stages does not yield better performance:
>
> | Sampling Iteration/$\rho$.         | SSIM↑ | PSNR↑  | LPIPS↓ |
> | ---------------------------------- | ----- | ------ | ------ |
> | [15k] / [0.1]                      | 0.817 | 23.313 | 0.242  |
> | [15k, 20k] / [0.2, 0.5]            | 0.817 | 23.44  | 0.247  |
> | [15k, 20k, 25k] / [0.4, 0.5, 0.5]  | 0.811 | 23.359 | 0.255  |
>
>
> - **Clarification on the interpretation in Figure 4.** We agree that the grass in Figure 4 (Bicycle) looks better before compaction, and we will revise our wording for clarity. The intended point is that our GHAP method is a post‑processing method: when the starting representation (here, **Atom‑GS**) lacks a strong densification strategy, compaction can further attenuate fine textures (e.g., grass), leading to a noticeable quality drop. In contrast, when GHAP is applied to a backbone with a more effective densification strategy (e.g., **Mini‑Splatting‑D**), it better preserves the visual quality on the Bicycle scene, as shown in Figure 8 of the supplementary material.
> - **Clarification on the comparison with random subsampling.** The random subsampling setting is not intended as a competitive baseline, but rather as an ablation to highlight the advantage of our structured compaction strategy. Its basic procedure is similar to GHAP: after training, it randomly removes a fixed percentage of Gaussians from the trained 3DGS model and then performs appearance optimization. We will clarify the purpose of this experiment and describe the procedure more explicitly in the revised manuscript to avoid confusion. We have included comparison with other baselines such as Mini‑Splatting during rebuttal and the results show that our method can achieve SOTA performance.
> - **Discussion on large Gaussian primitives that overlap with neighboring blocks.**  In our current formulation, we assign each Gaussian to a single block based on its mean. This is mainly due to the observation that the eigenvalues of the covariances are very small, hence have limited impact.
> - **Code release.** Due to rebuttal format constraints, we are unable to share a GitHub link at this stage. In the final version, we will release the source code along with example `.ply` files (before and after GHAP) and provide instructions for visualization and reproduction.
> - **Incorporating suggestions for paper writing.** We have added the number of Gaussian primitives in Table 2. We will reformat Algorithm 2 in the revised version to improve clarity and readability in the following ways: (i) restructuring the input/output section with a cleaner layout; (ii) refining indentation to better reflect the logical hierarchy of the steps; (iii) clarifying sub-step details and notations for improved understanding. We will also adjust the overall width and formatting to ensure the pseudocode fits well and is easy to follow.

---

> > ### Comment · Reviewer_28Ys · 2025-08-05
> >
> > Dear Authors,
> > Thank you for addressing all of my concerns!
> >
> > Reporting the runtime is very valuable and underlines that the method can be used in real life scenarios. I strongly suggest including them in the main paper. However, can you please elaborate why you claim that GHAP achieves the fastest runtime, although “Trimming the Fat” has a lower time in the provided table? Is there additional overhead introduced by “Trimming the Fat”?
> >
> > Providing further ablations also underlines the effectiveness of the method and demonstrates good evaluation.
> >
> > Another interesting comparison would be to apply the 10% method to MCMC with a maximum of 3M primitives and compare it to MCMC trained with a maximum of 300k primitives. This experiment is however fully optional and does not affect my final rating.
> >
> > Given that the authors have addressed all of my concerns (runtime, further comparisons, ablations, code release and formatting) and are willing to apply those changes in the final version, I will increase my rating to an accept.

---

> > > ### Author Response · Authors · 2025-08-05
> > >
> > > Dear reviewer, thank you for your thoughtful feedback and for recognizing our efforts in addressing your concerns. We sincerely appreciate your time and constructive suggestions, which have greatly improved our work.
> > >
> > > - We apologize for the oversight—this was a typo in our original response. To clarify, our method achieves the fastest runtime *among all compared approaches except* "Trimming the Fat," and we will correct this statement.
> > > - We fully agree that your suggested ablations strengthen the evaluation. Due to the time limit for discussion, we will try our best to run your suggested experiment on a simple dataset and we will include the full results in the final version.
> > > - We will incorporate all agreed-upon revisions—runtime analysis, expanded comparisons, ablations, and formatting—into the final manuscript. The code will also be released upon acceptance.
> > >
> > > Thank you again for your valuable input and for recommending acceptance. Please don’t hesitate to reach out if further clarifications would be helpful.

---

> > > ### Author Response · Authors · 2025-08-06
> > > **Follow-up for the optional experiment**
> > >
> > > > Another interesting comparison would be to apply the 10% method to MCMC with a maximum of 3M primitives and compare it to MCMC trained with a maximum of 300k primitives. This experiment is however fully optional and does not affect my final rating.
> > > >
> > >
> > > Dear reviewer,  Following your advice, we conducted the comparison on the *Tandt* dataset. Specifically, we trained two 3DGS-MCMC models: one with a maximum of 3M primitives and another with 300K primitives. We then applied our GHAP method to compact the 3M model down to 10% of its original size (resulting in 300K primitives). The results are summarized in the table below.
> > >
> > >
> > > | Method/dataset      | TNT    |         |        |             |
> > > |:--------------------|--------|---------|--------|-------------|
> > > |                     | SSIM↑  | PSNR↑   | LPIPS↓ | k Gaussians |
> > > | 3DGS-MCMC-300k      | 0.813  | 22.925  | 0.239  | 300         |
> > > | 3DGS-MCMC-3M + GHAP | 0.827  | 22.786  | 0.209  | 300         |
> > >
> > >
> > > Our findings demonstrate that the compacted model (**GHAP@3M→300K**) outperforms the directly trained 300K model (**MCMC@300K**) in terms of SSIM & LPIPS.

---

### Official Review · Reviewer_Cgrv · 2025-06-28

**Clarity:** 3
**Significance:** 2
**Originality:** 4
**Rating:** 4
**Confidence:** 4

**Summary:**

The paper proposes a method that makes 3D Gaussian Splatting representations compact.

The main pipeline is
- Start with an optimized 3D representation.
- Partition the 3D space using a KD-tree.
- Within each partition, run Gaussian Mixture Reduction (GMR) to decrease the number of Gaussians.
- Refine the appearance (colors) to obtain the final result.

The authors demonstrate that this pipeline can be applied on top of several existing methods, including the original 3DGS, Mini-Splatting, and AtomGS.

**Questions:**

- Would it be possible to include comparisons with a broader set of methods? This would highlight how effective the proposed GMR-based approach is relative to other techniques. Given this intriguing idea, I’d like to raise my initial score once those comparisons are more solid.
- The limitation section seems to be missing. Could you discuss the limitations during the rebuttal?

**Ethical Concerns:**

["NO or VERY MINOR ethics concerns only"]

**Final Justification:**

Common concerns about limited experimental results (including other baselines) and computational costs are well addressed.

**Limitations:**

Neither the main paper nor the supplementary material contains a Limitations section, which conflicts with the required checklist.

**Paper Formatting Concerns:**

No major issues.

**Quality:**

2

**Strengths And Weaknesses:**

**Strengths**
- Employing GMR is an interesting idea, and combining it with spatial partitioning via a KD-Tree effectively reduces computational cost.
- The storyline, from the optimal-transport to engineering details such as the KD-Tree, is easy to follow.


**Weaknesses**

- The experimental comparison is not thorough. Given the rapid advancements in this field, the paper should highlight its advantages more clearly. If the focus is post-training compactness, it should be compared with additional baselines such as MesonGS (ECCV 2024) and Fast Compression of 3D Gaussian Splatting (ICLR 2025). It would also be helpful to compare against end-to-end compact 3DGS variants such as ContextGS (NeurIPS 2024) and LocoGS (ICLR 2025).
- Rate–distortion (RD) curves are essential for comparison, yet Fig. 5 shows only the proposed method. Comparing RD curves to those of other methods would be helpful in demonstrating the effectiveness of the proposed method.

---

> ### Author Rebuttal · Authors · 2025-07-31
>
> Thank you for your thoughtful feedback. We're encouraged that you found our use of GMR interesting and the KD-tree partitioning effective in reducing computation. We also appreciate your recognition of the clarity and coherence of our presentation. Below, we provide point-by-point responses to your comments and will incorporate all of your suggestions into the final version.
>
> - **Additional results with suggested baselines.**  We have included five additional baselines: Trimming-the-Fat, Mini-Splatting, MesonGS, LocoGS, and 3DGS-MCMC. We excluded **Fast Compression of 3D Gaussian Splatting** from our comparisons because it is specifically tailored for post-processing compression based on properties of 3DGS, without aiming to compact the number of Gaussians. We also did not include **ContextGS**, as it does not provide the modified version of the diff-gaussian-rasterization module necessary for reproduction. We would like to emphasize that, for a fixed number of Gaussians, **MiniSplatting-D+GHAP** (our GMR pruning integrated into MiniSplatting) achieves the **best performance** among all methods. Our method combined with 3DGS also demonstrates **comparable results**. Importantly, the only difference between MiniSplatting and MiniSplatting-D+GHAP lies in the pruning step—our method replaces the default pruning with our proposed GMR-based approach. As shown in the results, this substitution significantly improves performance, thereby **highlighting the effectiveness of our proposed pruning strategy**.
>
> | Method-$\rho$/Dataset              | **TNT**     |              |             |             | **360**     |              |             |             | **DB**      |              |             |             |
> |-|-|-|-|-|-|-|-|-|-|-|-|-|
> |                                    | SSIM↑       | PSNR↑        | LPIPS↓      | k Guassians | SSIM↑       | PSNR↑        | LPIPS↓      | k Gaussians | SSIM↑       | PSNR↑        | LPIPS↓      | k Gaussians |
> | **3DGS+GHAP (ours)-10%**           | ***0.818*** | ***23.312*** | 0.242       | 157         | ***0.764*** | ***26.404*** | 0.314       | 263         | ***0.905*** | ***29.647*** | ***0.264*** | 248         |
> | Trimming the Fat-10%               | 0.776       | 21.535       | 0.293       | 156         | 0.731       | 25.255       | 0.343       | 263         | 0.887       | 28.056       | 0.302       | 247         |
> | MesonGS-10%                        | 0.811       | 20.714       | ***0.208*** | 157         | 0.773       | 24.924       | ***0.264*** | 263         | 0.896       | 28.693       | 0.264       | 248         |
> | 3DGS-MCMC-10%                      | 0.779       | 22.141       | 0.282       | 157         | 0.763       | 25.957       | 0.309       | 263         | 0.885       | 28.976       | 0.298       | 248         |
> | **3DGS+GHAP (ours)-20%**           | ***0.835*** | ***23.615*** | 0.212       | 314         | 0.788       | ***26.973*** | 0.275       | 527         | ***0.907*** | ***29.864*** | ***0.252*** | 496         |
> | Trimming the Fat-20%               | 0.819       | 22.498       | 0.232       | 315         | 0.781       | 26.494       | 0.28        | 524         | 0.9         | 29.082       | 0.272       | 494         |
> | MesonGS-20%                        | 0.822       | 20.699       | ***0.207*** | 314         | 0.776       | 25.006       | 0.262       | 527         | 0.897       | 28.696       | 0.262       | 496         |
> | 3DGS-MCMC-20%                      | 0.813       | 22.925       | 0.237       | 314         | ***0.801*** | 26.850       | ***0.257*** | 527         | 0.893       | 29.393       | 0.277       | 496         |
>
> | Method-$\rho$/Dataset              | **TNT**     |              |             |             | **360** |              |             |             | **DB**     |              |             |             |
> |-|-|-|-|-|-|-|-|-|-|-|-|-|
> |                                    | SSIM↑       | PSNR↑        | LPIPS↓      | k Guassians | SSIM↑           | PSNR↑        | LPIPS↓      | k Gaussians | SSIM↑      | PSNR↑        | LPIPS↓      | k Gaussians |
> | MiniSplatting-10%                  | 0.799       | 22.661       | 0.265       | 78          | 0.759           | 26.022       | 0.318       | 111         | 0.895      | 29.395       | 0.289       | 125         |
> | **MiniSplatting-D+GHAP-10% (ours)** | ***0.835*** | ***23.232*** | ***0.198*** | 79          | ***0.802***     | ***27.09***  | ***0.25***  | 112         | ***0.909*** | ***30.042*** | ***0.254*** | 127         |
> | MiniSplatting-20%                  | 0.824       | 22.953       | 0.223       | 142         | 0.794           | 26.728       | 0.267       | 215         | 0.904      | 29.763       | 0.265       | 240         |
> | **MiniSplatting-D+GHAP-20% (ours)** | ***0.855*** | ***23.403*** | ***0.171*** | 155         | ***0.821***     | ***27.31***  | ***0.214*** | 219         | ***0.912*** | ***30.17***  | ***0.238*** | 250         |
> | LocoGS                             | 0.843       | 23.655       | 0.191       | 571         | 0.798           | 27.049       | 0.257       | 673.8       | 0.903      | 29.972       | 0.261       | 529         |
>
> - **RD curve comparison.** Due to space constraints, we only include RD curves of PSNR for comparisons against LightGaussian, PUP‑3DGS, Trimming‑the‑Fat, and MesonGS on the Tanks&Temples datasets. These baselines were selected to ensure a fair comparison, as they apply post-compaction alone without incorporating other technical innovations. The results show that our method consistently outperforms these approaches across all cases.
>
> | $\rho$/Method | GHAP    | LightGaussian| PUP-3DGS  | Trimming the Fat| MesonGS |
> | :-------------------- |---------|--------------|-----------|-----------------|---------|
> | 10%                   | **23.312**  | 22.113       | 21.535    | 21.535          | 20.714  |
> | 20%                   | **23.615**  | 22.486       | 22.603    | 22.498          | 20.699  |
> | 30%                   | **23.740**  | 22.582       | 23.102    | 22.802          | 20.891  |
> | 40%                   | **23.656**  | 22.588       | 23.394    | 22.892          | 21.142  |
> | 50%                   | **23.791**  | 22.609       | 23.582    | 22.911          | 21.432  |
>
> - **Discussion on potential limitation of the method.** Thank you for raising this point. Our method's performance depends on the quality of the initial Gaussian representation. We plan to study different backbones and scene types (e.g., texture-heavy or far-field content), and explore perceptual losses to improve LPIPS fidelity under extreme retention.

---

> > ### Comment · Reviewer_Cgrv · 2025-08-05
> >
> > I'd like to thank authors for addressing raised concerns during the rebuttal.
> > Those concerns are properly addressed and I do not have any further questions.

---

> > > ### Author Response · Authors · 2025-08-05
> > >
> > > Dear reviewer, thank you for taking the time to review our work and for your valuable feedback during the review and rebuttal process. We sincerely appreciate your thoughtful comments and suggestions. All the points raised will be carefully addressed in the revised version of the manuscript.

---

### Official Review · Reviewer_D3B2 · 2025-07-01

**Clarity:** 2
**Significance:** 3
**Originality:** 3
**Rating:** 4
**Confidence:** 3

**Summary:**

The paper addresses the problem of reducing the number of Gaussians needed to represent a scene, using an optimal transport perspective. The authors first train a 3DGS model. The resulting Gaussians are then partitioned into blocks based on camera centers. In the first stage, the geometric parameters (i.e., centers and covariances) are iteratively adjusted using K-means to minimize the divergence between the original Gaussian mixture and the changed mixture. After this geometric optimization, the appearance parameters (color and opacity) are refined. The method is evaluated on three datasets, and the results show that, even when retaining only 10% of the Gaussians, the final rendering quality is largely preserved.

**Questions:**

As provided in the weakness section:

Please include timing results for your approach to help assess its computational cost.

Please add a comparison to [24] in Table 2, as it is also a relevant probabilistic method that performs well with fewer Gaussians.

Please revise Figures 3 and 4 for clarity and accuracy:
– In Figure 3, increase the font size or enlarge the image for better readability.
– In Figure 4, correct the "PNSR" typo (should be "PSNR"), verify the PSNR values (they appear to be switched), and consider adding a difference map to make the visual impact of your method more evident.

**Ethical Concerns:**

["NO or VERY MINOR ethics concerns only"]

**Final Justification:**

I have reviewed all of the reviewers’ comments, and overall, I believe that since the method outperforms all the other reported compression baselines, it represents a valuable contribution. I strongly encourage you to revise and correct the figures carefully in the final version of the paper. Additionally, please ensure that the new experiments and timing reports are incorporated, as they strengthen the evidence for your contribution.

**Limitations:**

- Please provide the limitations of your technique and possible future directions.

**Quality:**

3

**Strengths And Weaknesses:**

Strength:

Minimizing the number of Gaussians required to represent a scene is a relevant and timely research problem. The paper presents a novel approach that effectively reduces the number of Gaussians to 10% of the original set, while maintaining only marginal degradation in rendering quality.

Weakness:

The paper does not provide any information on the computational cost of applying the proposed method. Please include the runtime to help assess the practical overhead of integrating your method.

Parts of Figure 3 are difficult to read due to small font sizes. Consider enlarging the fonts or the entire figure for better clarity.

You cite [24] as another probabilistic approach that, while different in methodology, also performs well with fewer Gaussians compared to 3DGS. Please include this method in the comparisons (in Table 2), as it would be valuable to see how your approach performs compared to that.

Figure 4 needs correction. It contains a typo ("PNSR" instead of "PSNR") and the PSNR values appear to be mislabeled or inconsistent. Moreover, the visual differences between the “before” and “after” images are not clearly visible, even when zoomed in. It would be helpful to include a difference map to better highlight the changes introduced by your method, as they are not significant.

---

> ### Author Rebuttal · Authors · 2025-07-31
>
> Thank you for your constructive feedback. We’re encouraged that you view 3DGS compaction as a relevant and timely research problem and found our method both novel and effective. Below, we address your comments and will incorporate all of your suggestions into the final version.
>
> - **Computational cost of the method.**
>     - **Time complexity.** As described in L184, the overall time complexity of our method is $O(n\log n + \rho s^2\log n)$, where $\rho$ is the retention ratio, $n$ is the initial number of Gaussians, $s (\ll n)$ is the maximum number of Gaussians in each block. We use median splits to depth $d = \lfloor \log_2(n/s)\rfloor$, resulting in $2^{\text{depth}} = O(\log n)$ leaf blocks, each with at most $s$ Gaussians. Within each block, we reduce $s$ to $\rho s$ Gaussians. This gives $O(\rho s^2\log n)$  overall cost. Including KD‑tree construction $O(n\log n)$, the total time cost becomes $O(n\log n + \rho s^2\log n)$.
>     - **Memory complexity.** We will add the following analysis of memory consumption in the revision. The peak memory cost is $O(\rho s^2)$, dominated by the distance matrix in GMR. Global GMR requires $O(n \times \rho n)$ memory to store a full distance matrix; with our KD‑tree partitioning, the peak memory reduces to $O(s \times \rho s)$. In practice, batched distance evaluation or parallel processing further reduces workspace.
>     - **Empirical results.** Following your suggestion, we also provide empirical comparisons of runtime and memory usage with other post-processing compaction baselines in the table below. Our method achieves the fastest runtime with comparable memory consumption, resulting in negligible overhead when integrated into the rendering pipeline. Moreover, our method has the best PSNR.
>
> | Method/Retention Rate | GHAP      |        |        | LightGaussian |        |        | PUP-3DGS  |        |        | Trimming the Fat |        |        | MesonGS   |        |        |
> | :-------------------- | --------- | ------ | ------ | ------------- | ------ | ------ | --------- | ------ | ------ | ---------------- | ------ | ------ | --------- | ------ | ------ |
> |                       | time（s） | memory | PSNR↑  | time（s）      | memory   | PSNR↑  | time（s） | memory | PSNR↑  | time（s）        | memory | PSNR↑  | time（s） | memory | PSNR↑  |
> | 10%                   | 5.6       | 4857   | 23.312 | 10.8          | 3908   | 22.113 | 26.3      | 4001   | 21.535 | <0.1             | 3740   | 21.535 | 11.0      | 2830   | 20.714 |
> | 20%                   | 4.4       | 4895   | 23.615 | 11.2          | 3908   | 22.486 | 26.3      | 4001   | 22.603 | <0.1             | 3731   | 22.498 | 11.1      | 2830   | 20.699 |
> | 30%                   | 3.8       | 4976   | 23.740 | 9.0           | 3908   | 22.582 | 26.3      | 4001   | 23.102 | <0.1             | 3722   | 22.802 | 10.8      | 2830   | 20.891 |
> | 40%                   | 3.3       | 5060   | 23.656 | 10.7          | 3908   | 22.588 | 26.3      | 4001   | 23.394 | <0.1             | 3760   | 22.892 | 10.9      | 2830   | 21.142 |
> | 50%                   | 3.0       | 5141   | 23.791 | 10.8          | 3908   | 22.609 | 26.3      | 4001   | 23.582 | <0.1             | 3800   | 22.911 | 11.1      | 2830   | 21.432 |
>
> - **Comparison with 3DGS-MCMC[24].** In line with your suggestion, we include 3DGS-MCMC for a more comprehensive comparison. MCMC is an end-to-end 3DGS variant rather than a compaction method. Since it allows controlling the number of Gaussians, we also compare it with our method under the same Gaussian budget. As shown in the table, even compared to end-to-end approaches, our post-processing compaction method still achieves superior performance.
>
>
> | Method-$\rho$/Dataset    | **TNT**      |                |              |             | **360**     |              |               |             | **DB**      |              |             |             |
> |--------------------------|--------------|----------------|--------------|-------------|-------------| ------------ |---------------| ----------- |-------------| ------------ | ----------- | ----------- |
> |                          | SSIM↑        | PSNR↑          | LPIPS↓       | k Guassians | SSIM↑       | PSNR↑        | LPIPS↓        | k Gaussians | SSIM↑       | PSNR↑        | LPIPS↓      | k Gaussians |
> | **3DGS+GHAP (ours)-10%** | ***0.818***  | ***23.312***   | ***0.242***  | 157         | ***0.764*** | ***26.404*** | 0.314         | 263         | ***0.905*** | ***29.647*** | ***0.264*** | 248         |
> | 3DGS-MCMC-10%            | 0.779        | 22.141         | 0.282        | 157         | 0.763       | 25.957       | ***0.309***   | 263         | 0.885       | 28.976       | 0.298       | 248         |
> | **3DGS+GHAP (ours)-20%** | ***0.835***  | ***23.615***   | ***0.212***  | 314         | 0.788       | ***26.973*** | 0.275         | 527         | ***0.907*** | ***29.864*** | ***0.252*** | 496         |
> | 3DGS-MCMC-20%            | 0.813        | 22.925         | 0.237        | 314         | ***0.801*** | 26.850       | ***0.257***   | 527         | 0.893       | 29.393       | 0.277       | 496         |
>
>
> - **Potential Limitations and possible future directions.** Thanks for your suggestion. We will integrate the following potential limitations and future directions in the "Conclusion" section: First, the performance of our method depends on the quality of the initial Gaussian representation. We will further study backbones and scene types (e.g., texture‑heavy or far‑field content) and integrate perceptual losses to strengthen LPIPS‑oriented fidelity under extreme retention. Second, our block‑wise KD‑tree strategy brings strong scalability; going forward we plan to add overlap‑aware or multi‑scale partitioning to better handle large Gaussians spanning neighboring blocks and further improve efficiency. Third, we will move from fixed hyperparameters to auto‑tuned schedules guided by validation RD metrics and visibility statistics. Finally, future work also includes extending our method to dynamic 3DGS for real-time temporal scenes and developing adaptive reduction strategies for task-specific needs.
> - **Incorporating writing suggestions.** Thank you for your suggestion and careful observations. We will incorporate the writing suggestions and improve the readability of Figure 3 by enlarging the font sizes and adjusting the layout of the flowchart. We will also correct the typo in Figure 4. Regarding the PSNR values in Figure 4, we have confirmed that they are not switched. Unlike Table 2, which reports the average PSNR across views, Figure 4 presents the PSNR for a single representative view of each image. We will clarify this distinction in the final version of the paper. Additionally, we will include difference maps to better highlight subtle changes.
> ****

---

> > ### Comment · Reviewer_D3B2 · 2025-08-05
> >
> > "Regarding the PSNR values in Figure 4, we have confirmed that they are not switched". Are you implying that a PSNR of 0.92 is correct?!

---

> > > ### Comment · Reviewer_D3B2 · 2025-08-05
> > >
> > > Thank you for your detailed explanations. I have reviewed all of the reviewers’ comments, and overall, I believe that since your method outperforms all the other reported compression baselines, it represents a valuable contribution. I understand that figures cannot be uploaded here, but I strongly encourage you to revise and correct them carefully in the final version of the paper. Additionally, please ensure that the new experiments and timing reports are incorporated, as they strengthen the evidence for your contribution.

---

> > > > ### Author Response · Authors · 2025-08-05
> > > >
> > > > Dear reviewer,
> > > >
> > > > Thank you for your thoughtful review and positive assessment of our work. We sincerely appreciate your time and constructive feedback. We are glad to hear that you find our method’s performance and contribution valuable.
> > > >
> > > > As suggested, we will carefully revise and correct all figures in the final version of the paper to ensure clarity and accuracy. Additionally, we will incorporate the new experiments and timing reports to further strengthen the evidence supporting our results.
> > > >
> > > > Thank you again for your helpful comments, which have contributed to improving the clarity and quality of our work. We are committed to addressing all concerns thoroughly and improving the manuscript accordingly.

---

> > > > > ### Comment · Area_Chair_pUaD · 2025-08-08
> > > > >
> > > > > Dear reviewer,
> > > > >
> > > > > Please indicate whether you are satisfied with authors' response.
> > > > >
> > > > > And please ensure consistency of reviewers discussion with authors and reviewers final recommendation and final justification.
> > > > >
> > > > > AC

---

> > > ### Author Response · Authors · 2025-08-05
> > >
> > > Sorry we misunderstood your comment during the rebuttal process. You are absolutely right—the PSNR and SSIM values in the image were mistakenly swapped, and we appreciate your careful attention in identifying this error. We will correct this in the final version and ensure that all reported results are clearly and accurately presented.

---

### Official Review · Reviewer_1EBm · 2025-07-01

**Clarity:** 4
**Significance:** 4
**Originality:** 4
**Rating:** 4
**Confidence:** 4

**Summary:**

This paper presents Gaussian Herding Across Pens (GHAP), a Gaussian compaction method that integrates similar Gaussians, instead of pruning less-important Gaussians. The compaction pipeline consists of two stages, (a) geometric compaction via Gaussian Mixture Reduction (GMR) and (b) appearance finetuning. To enhance the efficiency, these compaction steps are conducted in a block-wise manner divided by KD-tree partition. Experimental results show that GHAP reduces the number of Gaussians of existing representations as a plug-and-play module. Also, it achieves comparable rendering quality with the same compaction setting compared to existing Gaussian compaction methods such as LightGaussian and PUP-3DGS.

**Questions:**

- In Table 2, the rendering quality of LightGaussian outperforms the rendering quality of this method for a 10% retention rate on Mip-NeRF 360 dataset. Mip-NeRF 360 dataset contains both indoor and outdoor scenes which have been employed as a standard real-world benchmark for novel-view synthesis. Can you explain why this method shows worse performance on Mip-NeRF 360 dataset?

- For better understanding, adding RD curves for Table 1 and Table 2 would be helpful.

- Although several compaction methods are mentioned in Section 2, the authors provide a comparison only with LightGaussian and PUP-3DGS. It is recommended to conduct an additional comparison with RadSplat, Mini-Splatting, and Trimming-the-Fat.

**Ethical Concerns:**

["NO or VERY MINOR ethics concerns only"]

**Final Justification:**

During the rebuttal period, the authors have addressed all of my concerns regarding the limited technical advancements and performance. Notably, they have demonstrated that the proposed method outperforms various approaches through extensive experiments. Therefore, I raise my final rating to positive.

**Limitations:**

- Despite the effectiveness of the proposed compaction strategy, this paper contains only limited technical contributions to be accepted at this conference.

- Although the experimental results show comparable performance, they are insufficient to clearly demonstrate that the proposed method outperforms existing approaches.

**Paper Formatting Concerns:**

There are no paper formatting concerns.

**Quality:**

3

**Strengths And Weaknesses:**

### Strengths

- While the existing compaction methods reduce the number of Gaussians by pruning, this method integrates similar Gaussians.
- This method can be adapted to existing Gaussian representations as a plug-and-play module.
- Experimental results indicate that this method can maintain high fidelity despite a small retention rate.

---

### Weaknesses

- To validate the effectiveness of the KD-tree partitioning strategy, it would be recommended to provide an ablation study on it.

- The technical contribution of this paper is a simple compaction method for 3D Gaussians, which may be insufficient to be accepted at this conference.

- The experimental results show less effective results according to the experimental setting. To be specific, it cannot outperform all existing compaction methods for all datasets. Despite the high fidelity, comparative methods show better scores for some experimental settings, as shown in Table 2.

---

> ### Author Rebuttal · Authors · 2025-07-31
>
> Thanks for giving us excellent scores on originality and significance. We respond to your comments below and will
> incorporate all of your suggestions into the final version.
>
> - **Highlight of our technical contribution.** We respectfully disagree with the assessment that the technical contribution of our method is limited. We clarify and highlight the key innovations of our work:
>     - **Novel perspective.** We are the first to reinterpret Gaussian primitives in 3DGS as components of a **Gaussian mixture**, using their density functions to model scene geometry. This contrasts with prior compaction methods that ignore geometric structure and often produce distortions (see Fig. 2), whereas our approach preserves **geometric fidelity**, offering a new and impactful direction for 3DGS.
>     - **Adaptation of GMR to 3DGS.** We are the first to adapt GMR to 3DGS. We introduce a **novel cost function** that yields **closed-form, low-cost updates**. We also develop a **block-wise GMR algorithm guided by a KD-tree**, enabling efficient large‑scale scene compaction. These strategies are non-trivial and bridges theory with practical scalability.
>     - **Plug & play design.** Our method is **post-hoc** and **compatible** with any existing 3DGS pipeline, making it highly practical and broadly applicable. With minimal overhead, our approach achieves **SOTA compaction performance**, both in quality and efficiency.
> - **Clarification on Table 2.** During the rebuttal process, we found a configuration error in the LightGaussian baseline: it was mistakenly retrained for an additional 30k steps without pruning, exceeding the intended 5k-step budget. This led to significantly more Gaussians (e.g., 1.6M vs. 157k on average), and an unfair comparison. We have corrected the setup and present the updated results below. Our method achieves superior overall performance. We thank the reviewer and apologize for the oversight.
>
> | Method-$\rho$/Dataset       | **TNT**     |              |             |             | **360** |              |          |             | **DB**      |              |             |             |
> | - |-| - | -| - | - | - |-| - |-------------| - |-| - |
> |                             | SSIM↑       | PSNR↑        | LPIPS↓      | k Guassians | SSIM↑           | PSNR↑        | LPIPS↓   | k Gaussians | SSIM↑       | PSNR↑        | LPIPS↓      | k Gaussians |
> | **GHAP (ours)-10%**         | ***0.818*** | ***23.312*** | ***0.242*** | 157         | ***0.764***     | ***26.404*** | ***0.314*** | 263         | ***0.905*** | ***29.647*** | ***0.264*** | 248         |
> | LightGaussian-10%           | 0.756       | 22.113       | 0.306       | 158         | 0.735           | 25.674       | 0.331    | 263         | 0.869       | 28.01        | 0.327       | 248         |
> | **GHAP (ours)-20%**         | ***0.835*** | ***23.615*** | ***0.212*** | 314         | ***0.788***     | ***26.973*** | ***0.275*** | 527         | ***0.907*** | ***29.864*** | ***0.252*** | 496         |
> | LightGaussian-20%           | 0.779       | 22.486       | 0.271       | 315         | 0.765           | 26.353       | 0.288    | 526         | 0.873       | 28.011       | 0.315       | 495         |
> - **Comparison with baselines.** We have included 5 additional baselines: Trimming-the-Fat, Mini-Splatting, MesonGS, LocoGS, and 3DGS-MCMC. RadSplat is excluded due to unavailability of public code. For a fixed number of Gaussians, **MiniSplatting-D+GHAP** (MiniSplatting with our pruning) achieves the best performance. Notably, the only difference from MiniSplatting is the pruning step—replacing it with our method yields significant gains, underscoring the **effectiveness of our GMR strategy**.
>
> | Method-$\rho$/Dataset              | **TNT**     |              |             |             | **360**     |              |             |             | **DB**      |              |             |             |
> |-|-|-|-|-|-|-|-|-|-|-|-|-|
> |                                    | SSIM↑       | PSNR↑        | LPIPS↓      | k Guassians | SSIM↑       | PSNR↑        | LPIPS↓      | k Gaussians | SSIM↑       | PSNR↑        | LPIPS↓      | k Gaussians |
> | **3DGS+GHAP (ours)-10%**           | ***0.818*** | ***23.312*** | 0.242       | 157         | ***0.764*** | ***26.404*** | 0.314       | 263         | ***0.905*** | ***29.647*** | ***0.264*** | 248         |
> | Trimming the Fat-10%               | 0.776       | 21.535       | 0.293       | 156         | 0.731       | 25.255       | 0.343       | 263         | 0.887       | 28.056       | 0.302       | 247         |
> | MesonGS-10%                        | 0.811       | 20.714       | ***0.208*** | 157         | 0.773       | 24.924       | ***0.264*** | 263         | 0.896       | 28.693       | 0.264       | 248         |
> | 3DGS-MCMC-10%                      | 0.779       | 22.141       | 0.282       | 157         | 0.763       | 25.957       | 0.309       | 263         | 0.885       | 28.976       | 0.298       | 248         |
> | **3DGS+GHAP (ours)-20%**           | ***0.835*** | ***23.615*** | 0.212       | 314         | 0.788       | ***26.973*** | 0.275       | 527         | ***0.907*** | ***29.864*** | ***0.252*** | 496         |
> | Trimming the Fat-20%               | 0.819       | 22.498       | 0.232       | 315         | 0.781       | 26.494       | 0.28        | 524         | 0.9         | 29.082       | 0.272       | 494         |
> | MesonGS-20%                        | 0.822       | 20.699       | ***0.207*** | 314         | 0.776       | 25.006       | 0.262       | 527         | 0.897       | 28.696       | 0.262       | 496         |
> | 3DGS-MCMC-20%                      | 0.813       | 22.925       | 0.237       | 314         | ***0.801*** | 26.850       | ***0.257*** | 527         | 0.893       | 29.393       | 0.277       | 496         |
>
> | Method-$\rho$/Dataset              | **TNT**     |              |             |             | **360** |              |             |             | **DB**     |              |             |             |
> |-|-|-|-|-|-|-|-|-|-|-|-|-|
> |                                    | SSIM↑       | PSNR↑        | LPIPS↓      | k Guassians | SSIM↑           | PSNR↑        | LPIPS↓      | k Gaussians | SSIM↑      | PSNR↑        | LPIPS↓      | k Gaussians |
> | MiniSplatting-10%                  | 0.799       | 22.661       | 0.265       | 78          | 0.759           | 26.022       | 0.318       | 111         | 0.895      | 29.395       | 0.289       | 125         |
> | **MiniSplatting-D+GHAP-10% (ours)** | ***0.835*** | ***23.232*** | ***0.198*** | 79          | ***0.802***     | ***27.09***  | ***0.25***  | 112         | ***0.909*** | ***30.042*** | ***0.254*** | 127         |
> | MiniSplatting-20%                  | 0.824       | 22.953       | 0.223       | 142         | 0.794           | 26.728       | 0.267       | 215         | 0.904      | 29.763       | 0.265       | 240         |
> | **MiniSplatting-D+GHAP-20% (ours)** | ***0.855*** | ***23.403*** | ***0.171*** | 155         | ***0.821***     | ***27.31***  | ***0.214*** | 219         | ***0.912*** | ***30.17***  | ***0.238*** | 250         |
> | LocoGS                             | 0.843       | 23.655       | 0.191       | 571         | 0.798           | 27.049       | 0.257       | 673.8       | 0.903      | 29.972       | 0.261       | 529         |
>
>
> - **Effectiveness of KD-tree partition.** On the NeRF-synthetic-mic scene, our ablation shows that increasing KD-tree depth reduces memory and runtime, while PSNR first improves then drops. This suggests finer partitions help GMR compact regions more effectively, but overly fine splits may fragment primitives and hurt quality.
>
> | Depth | 0      |            |            | 1       |           |            | 2       |            |            | 3      |            |            | 4       |            |            | 5          |            |            | 6      |            |            | 7       |            |            | 8      |            |            |
> | - |-|-| - |-|-|-|-|-|-|-|-|-|-|-|-|-|-|-|-|-|-|-|-|-|-|-|-|
> |              | PSNR↑  | Runtime(s) | Memery(MB) | PSNR↑   | Runtime(s) | Memery(MB) | PSNR↑   | Runtime(s) | Memery(MB) | PSNR↑  | Runtime(s) | Memery(MB) | PSNR↑   | Runtime(s) | Memery(MB) | PSNR↑      | Runtime(s) | Memery(MB) | PSNR↑  | Runtime(s) | Memery(MB) | PSNR↑   | Runtime(s) | Memery(MB) | PSNR↑  | Runtime(s) | Memery(MB) |
> | 3DGS+GHAP    | 13.610 | 2.7        | 3706       | 14.427  | 5.0       | 3247       | 14.908  | 6.5        | 3134       | 14.756 | 7.0        | 3106       | 16.138  | 5.8        | 3098       | 16.376     | 5.1        | 3097       | 15.440 | 4.5        | 3096       | 15.7963 | 3.7        | 3095       | 15.488 | 1.0        | 3095       |
>
> - **RD curve.** We compare PSNR RD curves on Tanks&Temples against LightGaussian, PUP‑3DGS, Trimming‑the‑Fat, and MesonGS-baselines using post-compaction only for fair comparison. Our method consistently outperforms them across all cases.
>
> | Method/$\rho$ | GHAP    | LightGaussian| PUP-3DGS  | Trimming the Fat| MesonGS |
> | - |-|-|-|-|-|
> | 10%                   | **23.312**  | 22.113       | 21.535    | 21.535          | 20.714  |
> | 20%                   | **23.615**  | 22.486       | 22.603    | 22.498          | 20.699  |
> | 30%                   | **23.740**  | 22.582       | 23.102    | 22.802          | 20.891  |
> | 40%                   | **23.656**  | 22.588       | 23.394    | 22.892          | 21.142  |
> | 50%                   | **23.791**  | 22.609       | 23.582    | 22.911          | 21.432  |

---

> > ### Comment · Reviewer_1EBm · 2025-08-05
> >
> > I appreciate the authors’ efforts during the rebuttal period. Especially, highlighting the technical advancements helps me better understand the contributions of this work. Despite the rebuttal, I have several remaining concerns.
> >
> > ---
> >
> > **Comparison to LightGaussian**
> >
> > The experimental setup for LightGaussian needs clarification. If no pruning is performed, there should be no reason for its rendering quality to drop compared to the original 3DGS backbone, as shown in Table 2 of the submission. However, the results show a noticeable performance drop. This indicates that additional compression techniques, such as quantization, may have been applied. Could the authors clarify whether any compression techniques other than pruning are used for LightGaussian? For a fair comparison, all methods should apply only pruning.
> >
> > ---
> >
> > **Comparison to Mini-Splatting**
> >
> > It seems that different backbones are used for Mini-Splatting. I understand that GHAP is applied to Mini-Splatting-D, whereas it is compared to Mini-Splatting (not the D version). Since Mini-Splatting-D employs a depth-based densification strategy to enhance rendering quality, it is not a fair comparison to evaluate Mini-Splatting and Mini-Splatting-D + GHAP. Please clarify this point.
> >
> > ---
> >
> > **Optimization steps**
> >
> > Several methods (3DGS-MCMC, Mini-Splatting, and LocoGS) optimize Gaussians for 30K iterations by default, whereas this method requires additional 15K iterations for fine-tuning. For a fair comparison, the total number of optimization steps for each method should be the same. Could the authors clarify the optimization steps used in the provided results?

---

> > > ### Author Response · Authors · 2025-08-05
> > >
> > > **Comparison to LightGaussian.** Thank you for your attention to this detail. In our latest LightGaussian experiments, we used **only pruning**—no quantization was applied. This was accomplished through the official LightGaussian implementation, which offers a dedicated interface for pruning-only functionality. During the rebuttal phase, we updated our experimental results to reflect this pruning-only setup, ensuring consistency in our reported findings.
> > >
> > > ---
> > >
> > > **Clarification on comparison to Mini-Splatting.**  Sorry for any confusion regarding the comparison between Mini-Splatting (MS) and Mini-Splatting-D + GHAP (MSD+GHAP). To clarify, the experimental settings for both methods are as follows:
> > >
> > > 1. **Shared Backbone & training (First 15K iterations)**: Both MS and MSD+GHAP use the same backbone and follow the default MS training pipeline for the first 15K iterations.
> > > 2. **Compaction at 15001-th iteration**:
> > >     - MS applies its own **score-based pruning** to reduce the number of Gaussians to *ρ*%.
> > >     - MSD+GHAP instead uses our proposed **compaction method** to achieve the same *ρ*% reduction.
> > > 3. **Fine-Tuning (Next 15K iterations)**: Both methods employ the same fine-tuning strategy. The only minor difference is that MS further removes a small number of outliers (0.1%) during fine-tuning, resulting in a slight additional reduction in Gaussians.
> > >
> > > This ensures a fair comparison, with the key distinction being our compaction method versus MS’s pruning at the 15K-iteration mark.
> > >
> > > ---
> > >
> > > **Clarification on optimization steps.**
> > >
> > > To ensure a fair comparison in our experiments, all methods undergo **30,000 total iterations** under consistent training conditions. MCMC uses 30K iterations of its own update process in  Kheradmand et al. [NeurIPS, 2024]. For other baselines, we summarize the backbone architecture, compaction methods, and fine-tuning steps in the table below.
> > >
> > >
> > > | Method               | Backbone (0-15k iterations) | Compaction (15001th iteration) | Fine-tune (150001-30K iterations) |
> > > |----------------------|-----------------------------|--------------------------------|-----------------------------------|
> > > | 3DGS                 | [a]                         | None                           | [d]                               |
> > > | 3DGS+GHAP            | [a]                         | Our compaction                 | [d]                               |
> > > | LightGaussian        | [a]                         | LightGaussian compaction       | [d]                               |
> > > | PUP-3DGS             | [a]                         | PUP-3DGS compaction            | [d]                               |
> > > | Trimming the FAT     | [a]                         | Trimming the FAT compaction    | [d]                               |
> > > | MesonGS              | [a]                         | MesonGS compaction             | [d]                               |
> > > | MiniSplatting        | [b]                         | MiniSplatting compaction       | [d]                               |
> > > | MiniSplatting-D+GHAP | [b]                         | Our compaction                 | [d]                               |
> > > | LocoGS               | [c]                         | LocoGS compaction              | [d]                               |
> > >
> > >
> > > [a] Vanilla 3DGS densification and pruning in Kerbl et al. [ACM Trans. Graph., 2023]
> > >
> > > [b] Mini-Splatting densification and pruning in Fang & Wang [ECCV, 2024]
> > >
> > > [c] LocoGS 3DGS update in Shin et al. [ICLR, 2025]
> > >
> > > [d] 3DGS fine-tuning in Kerbl et al. [ACM Trans. Graph., 2023]
> > >
> > > **Key implementation details:**
> > >
> > > - For compaction methods applicable to pre-trained models (e.g., LightGaussian, PUP-3DGS, Trimming the FAT, MesonGS), we initialize from the **same backbone model** (trained using vanilla 3DGS) and apply their respective compaction directly—**excluding any compression modules**.
> > > - All methods undergo **identical fine-tuning (15k iterations)** to achieve the target retention ratio.
> > >
> > > This standardized protocol ensures that performance differences stem solely from the methods themselves, eliminating variations due to training procedures.

---

> > > > ### Comment · Reviewer_1EBm · 2025-08-06
> > > >
> > > > Thank you for the response. I recommend including all of these details in the final manuscript to improve clarity. As all of my concerns have been addressed, I will update my rating to positive.

---

> > > > > ### Author Response · Authors · 2025-08-06
> > > > >
> > > > > Dear reviewer,
> > > > >
> > > > > Thank you for your thoughtful review and for your constructive feedback. We sincerely appreciate your time and effort to evaluate our work.
> > > > >
> > > > > We are pleased to hear that your concerns have been addressed, and we greatly appreciate your recommendation to include additional details in the final manuscript to enhance clarity. We will carefully incorporate all of your suggestions in the revised version.
> > > > >
> > > > > Thank you again for your valuable input and for updating your rating.

---

### Comment · Area_Chair_pUaD · 2025-08-04

Dear reviewers,

Please post your first response as soon as possible, so there is time for back and forth discussion with the authors.

All reviewers should respond to the authors, so that the authors know their rebuttal has been read.

Thanks,

AC

---

### Note · Authors · 2025-08-13

We sincerely thank all reviewers for their constructive feedback, and the AC for active coordination throughout the discussion phase. The discussion between authors and reviewers confirms that our rebuttal effectively addressed the reviewers’ questions. Below, we summarize the key points from our rebuttal.

- **Novelty & contribution clarification:** Our work is the first to interpret Gaussian primitives in 3DGS as components of a Gaussian mixture, introducing a novel, scalable Gaussian Mixture Reduction algorithm and a plug-and-play design that enables efficient, high-quality post-hoc scene compaction.
- **Comprehensive baseline evaluation:** We expanded the comparisons to include five additional baselines—MesonGS, Trimming-the-Fat, Mini-Splatting, LocoGS, and 3DGS-MCMC—covering both post-hoc compaction and end-to-end methods. Results across multiple datasets and RD curve analysis consistently show GHAP achieves the highest or competitive PSNR/SSIM with low LPIPS at aggressive compaction rates.
- **Runtime & memory efficiency:** We provided both theoretical complexity analyses and empirical benchmarks, demonstrating that GHAP achieves the fastest runtime among practical compaction methods (except Trimming the Fat) while introducing negligible memory overhead in the rendering pipeline.
- **Ablations & design justification:** Additional experiments validate our loss function and cost design choices (LPIPS-loss variants, opacity-term weighting, joint geometry–appearance tuning) and confirm the efficiency–quality trade-off via KD-tree depth ablations. These results further support the robustness and adaptability of our approach.

**Commitment:** All updated experiments, clarifications, and the limitations section will be carefully incorporated into the revised version.

---

### Decision · Program_Chairs · 2025-09-17

**Decision:**

Accept (spotlight)

**Comment:**

This paper proposes to view 3DGS representations as a Gaussian mixture and perform compaction from the perspective of Gaussian mixture reduction via optimal transport.
Strength:
Framework is novel (1EBm, D3B2, Cgrv), plug-and-play (1EBm), high performance (1EBm, D3B2, 28Ys).
Weakness:
Ablation study (1EBm, 28Ys), effectiveness (1EBm), technical novelty (1EBm),  computation (D3B2, Cgrv, 28Ys), baseline comparison (D3B2, Cgrv, 28Ys)

After rebuttal, this paper receives all positive ratings (5444). Most reviewers think the proposed method is novel and the performance is high. Although some common concerns regarding computation cost, ablation study anb baseline comparisons are raised, the authors have done a good job during the rebuttal process, and all the reviewers are satisfied and vote for acception. Thus, the AC recommend to accept this paper. Please do include those new results and discussions in later versions.